

# Building the COllaborative Carbon Column Observing Network (COCCON): Long term stability and ensemble performance of the EM27/SUN Fourier transform spectrometer

Matthias Frey[1], Mahesh Kumar Sha[2,a], Frank Hase[1], Matthäus Kiel[3,a], Thomas Blumenstock[1], Roland Harig[4], Gregor Surawicz[4], Nicholas M. Deutscher[5], Kei Shiomi[6], Jonathan Franklin[7], Hartmut Bösch[8,9], Jia Chen[10], Michel Grutter[11], Hirofumi Ohyama[12], Youwen Sun[13], André Butz[14,b], Gizaw Mengistu Tsidu[15], Dragos Ene[16], Debra Wunch[17], Zhensong Cao[13], Omaira Garcia[18], Michel Ramonet[19], Felix Vogel[20], and Johannes Orphal[1]

[1]Karlsruhe Institute of Technology (KIT), Institute for Meteorology and Climate Research (IMK-ASF), Karlsruhe, Germany
[2]Royal Belgian Institute for Space Aeronomy, Brussels, Belgium
[3]Division of Geological and Planetary Sciences, California Institute of Technology, Pasadena, CA, USA
[4]Bruker Optics GmbH, Ettlingen, Germany
[5]Centre for Atmospheric Chemistry, School of Chemistry, University of Wollongong, Wollongong, Australia
[6]Japan Aerospace Exploration Agency, Tsukuba, Japan
[7]Harvard University, Cambridge, USA
[8]Department of Physics and Astronomy, University of Leicester, UK
[9]National Centre for Earth Observation (NCEO), University of Leicester, UK
[10]Technische Universität München, Munich, Germany
[11]Universidad National Autonoma de Mexico, Mexico City, Mexico
[12]National institute for Environmental Studies, Tsukuba, Japan
[13]Anhui Institute of Optics and Fine Mechanics, Hefei, China
[14]Institut für Umweltphysik, Universität Heidelberg, Germany
[15]Botswana International University of Science and Technology, Gaborone, Botswana
[16]National Institute for Research and Development in Optoelectronics, Magurele, Romania
[17]University of Toronto, Toronto, Canada
[18]Izaña Atmospheric Research Centre (IARC), Meteorological State Agency of Spain (AEMET), Tenerife, Spain
[19]Laboratoire des sciences du climat et de l'environnement, Gif-Sur-Yvette, France
[20]Environment Canada, Toronto, Canada
[a]before at: Karlsruhe Institute of Technology (KIT), Institute for Meteorology and Climate Research (IMK-ASF), Karlsruhe, Germany
[b]before at: Institut für Physik der Atmosphäre,Deutsches Zentrum für Luft- und Raumfahrt e. V., Oberpfaffenhofen, Germany

**Correspondence:** M. Frey (m.frey@kit.edu)

**Abstract.** In a 3.5 year long study, the long term performance of a mobile Bruker EM27/SUN spectrometer, used for greenhouse gases observations, is checked with respect to a co-located reference Bruker IFS 125HR spectrometer, which is part of the Total Carbon Column Observing Network (TCCON). We find that the EM27/SUN is stable on timescales of several years, qualifying it as an useful supplement for the existing TCCON network in remote areas. For achieving consistent performance, such an extension requires careful testing of any spectrometers involved by application of common quality assurance measures. One major aim of the COllaborative Carbon Column Observing Network (COCCON) infrastructure is to provide





these services to all EM27/SUN operators. In the framework of COCCON development, the performance of an ensemble of 30 EM27/SUN spectrometers was tested and found to be very uniform, enhanced by the centralized inspection performed at the Karlsruhe Institute of Technology prior to deployment. Taking into account measured instrumental line shape parameters for each spectrometer, the resulting average bias across the ensemble in $XCO_2$ is 0.20 ppm, while it is 0.8 ppb for $XCH_4$. As

indicated by the executed long-term study on one device presented here, the remaining empirical calibration factor deduced for each individual instrument can be assumed constant over time. Therefore the application of these empirical factors is expected to further improve the EM27/SUN network conformity beyond the raw residual bias reported above.

# 1 Introduction

Precise measurements of atmospheric abundances of greenhouse gases (GHGs), especially carbon dioxide ($CO_2$) and methane

($CH_4$), are of utmost importance for the estimation of emission strengths and flux changes (Olsen and Randerson, 2004). Furthermore these measurements can be directly used to evaluate emissions reductions as demanded by international treaties, e.g. the Paris COP21 agreement (http://unfccc.int/resource/docs/2015/cop21/eng/l09r01.pdf/). The Total Carbon Column Observing Network (TCCON) measures total columns of $CO_2$ and $CH_4$ with reference precision (Wunch et al., 2011), however the instruments used by this network are rather expensive and need large infrastructure to be set up and expert maintenance.

Therefore TCCON stations have sparse global coverage, especially in Africa, South America and large parts of Asia (Wunch et al., 2015). Current satellites like the Orbiting Carbon Observatory-2 (OCO-2) (Frankenberg et al., 2015) and the Greenhouse Gases Observing Satellite (GOSAT) (Morino et al., 2011) on the other hand offer global coverage. Nonetheless, they suffer from coarse temporal resolution (the repeat cycle of OCO-2 is 16 days), and in the case of GOSAT from sparse spatial sampling as well as limited precision of a single measurement. These limitations mostly inhibit a straightforward estimation of

the emission strength of localised sources of $CO_2$ and $CH_4$ like cities, landfills, swamps or fracking and mining areas from satellite observations. However, recently OCO-2 data was used for estimating the source strength of power plants (Nassar et al., 2017).

The previously described Bruker EM27/SUN portable FTIR spectrometer (Gisi et al., 2011; Frey et al., 2015; Hedelius et al., 2016) is a promising instrument to overcome the above mentioned shortcomings as it is a mobile, reliable, easy to deploy and

25 low-cost supplement to the Bruker IFS 125HR spectrometer used in the TCCON network. So far the EM27/SUN was mainly used in campaigns for the quantification of local sinks and sources (Hase et al., 2015; Chen et al., 2016). In this work the long term performance of the EM27/SUN with respect to a reference high resolution TCCON instrument is investigated. Additionally, the ensemble performance of several EM27/SUN spectrometers is tested. During 2014-2018, 30 EM27/SUN were calibrated at the Karlsruhe Institute of Technology (KIT) before being shipped to the customers. Several instruments that were

30 distributed before this calibration routine at KIT was established, were upgraded with a second channel for CO observations at Bruker Optics$^{TM}$ and after this also calibrated at KIT. This results in a unique data set as all EM27/SUN are directly calibrated with respect to a reference EM27/SUN, continuously operated at KIT, as well as a co-located TCCON instrument. From this data set an EM27/SUN network precision and accuracy can be estimated.



## 2 Methodology

### 2.1 TCCON data set

As part of the TCCON, the Karlsruhe Institute of Technology (KIT) operates a high resolution ground based spectrometer at KIT, Campus North (CN) near Karlsruhe (49.100°N, 8.439°E, 112 m a.s.l.). Standard TCCON instruments have been

described in great detail elsewhere (Washenfelder et al., 2006; Wunch et al., 2011). The Karlsruhe instrument, in the following called HR125, is the first demonstration of synchronized recordings of TCCON near infrared (NIR) and NDACC mid infrared (MIR) spectra using a dedicated dichroic beamsplitter (BS) arrangement (Optics Balzers Jena GmbH, Germany) with a cut-off wavenumber of 5250 $cm^{-1}$. It uses an InGaAs (indium gallium arsenide) detector in conjunction with an InSb (indium antimonide) detector, details can be found in Kiel et al. (2016b). By the TCCON measurements, the relevant wavenumber

region 4000 - 11000 $cm^{-1}$ is covered so that, among other species, $O_2$, $CO_2$, $CH_4$, CO and $H_2O$ can be retrieved. The TCCON measurements were chosen as reference measurements because these gases are also measured by the EM27/SUN spectrometer. For TCCON measurements in the NIR the HR125 records single sided interferograms with a resolution of 0.014 $cm^{-1}$ or 0.0075 $cm^{-1}$, corresponding to a maximum optical path difference (MOPD) of 64 cm and 120 cm. The recording time for a typical measurement consisting of two forward and two backward scans is 212 s, and 388 s, respectively. The applied

scanner velocity is 20 kHz. The TCCON site Karlsruhe participated in the IMECC aircraft campaign (Messerschmidt et al., 2011; Geibel et al., 2012). The spectrometer has been used for calibrating all gas cells used by TCCON for instrumental line shape (ILS) monitoring (Hase et al., 2013).

TCCON data processing is performed using the GGG Suite software package (Wunch et al., 2011). In this study, the current release version, GGG 2014 is used (Wunch et al., 2015). The software package includes a pre-processor correcting for solar

brightness fluctuations (Keppel-Aleks et al., 2007) and performing a fast Fourier transform including phase error correction routine to convert recorded interferograms into solar absorption spectra. Note that forward and backward scans are split by the preprocessing software and analyzed separately. The central part of the software package is the non-linear least-squares retrieval algorithm GFIT. It performs a scaling retrieval with respect to an a priori profile, then integrates the scaled profile over height to calculate the total column of the gas of interest. The software package additionally uses meteorological data (NCEP)

and provides daily a priori gas profiles. TCCON converts the retrieved total column abundances $VC_{gas}$ of the measured gases into column-averaged dry air mole fractions (DMFs), where the DMF of a gas is denoted $X_{gas} = \frac{VC_{gas}}{VC_{O_2}} \times 0.2095$. In this representation several errors cancel out that affect both the target gas and $O_2$. However, residual bias with respect to in situ measurements still persists, as well as a residual spurious dependence of retrieval results on the apparent airmass. Therefore the GGG suite also includes a post-processing routine applying an empirical airmass-dependent correction factor (ADCF)

and airmass-independent correction factor (AICF). The AICF are deduced from comparisons with in situ instrumentation on aircrafts (Wunch et al., 2010).





## 2.2 HR125 low resolution data set

In addition to the afore mentioned TCCON data product, a second data product from the HR125 will be used in this work, in the following called HR125 LR. For this product the raw interferograms are first truncated to the resolution of the EM27/SUN, 0.5 $cm^{-1}$. At 0.5 $cm^{-1}$ resolution, the ILS of the HR125 is expected to be nearly nominal. However, to avoid any systematic bias

of the HR125 LR data with respect to the EM27/SUN results, the same procedure for ILS determination from $H_2O$ signatures in open path lab air spectra was applied and the resulting ILS parameters adopted for the trace gas analysis. The analysis procedure will be explained in detail in section 2.3. The reason for the construction of this HR125 LR data set is that with this approach the analysis for the two instruments can be performed exactly the same way. The resolution is harmonized, the averaging kernels for a given airmass are nearly identical. Differences between the EM27/SUN and the HR125 LR data set

can then be attributed to instrumental features alone and do not need to be disentangled from retrieval software, resolution and airmass dependency differences. Note that for the low resolution data set, forward and backward scans are averaged and then analysed whereas they are analysed separately for the TCCON data set. Therefore number of coincident measurements with the EM27/SUN data set compared to the TCCON data set is lower.

## 2.3 EM27/SUN data set

The EM27/SUN spectrometer, which was developed by KIT in collaboration with Bruker Optics$^{TM}$, is utilized for the acquisition of solar spectra. The instrument has been described in great detail in Gisi et al. (2012), in the following a short overview is given. Central part of this Fourier transform spectrometer (FTS) is a RockSolid$^{TM}$ pendulum interferometer with two cube corner mirrors and a $CaF_2$ beamsplitter. The EM27/SUN routinely records double sided interferograms, the compensated BS design minimizes the curvature in the phase spectrum. This setup achieves high stability against thermal influences

and vibrations. The retroreflectors are gimbal-mounted, which results in frictionless and wear-free movement. In this aspect the EM27/SUN is more stable than the HR125 high resolution FTS, which suffers from wear because of the use of friction bearings on the moving retroreflector. Over time this leads to shear misalignment and requires regular realignment (Hase, 2012). The gimbal-mounted retroreflectors move a geometrical distance of 0.45 cm leading to an optical path difference of 1.8 cm which corresponds to a spectral resolution of 0.5 $cm^{-1}$.

In a first pre-processing step, a solar brightness fluctuation correction is performed similar to Keppel-Aleks et al. (2007). Furthermore, the recorded interferograms are Fourier transformed using the Norton-Beer-Medium apodisation function (Davis et al., 2010). This apodisation is useful for reducing sidelobes around the spectral lines, an undesired feature in low resolution spectra, which would complicate the further analysis. A quality control, which filters interferograms with intensity fluctuations above 10 % and intensities below 10 % of the maximal signal range, is also applied.

In this work, spectra were analyzed utilizing the PROFFIT Version 9.6, a non-linear least-squares spectral fitting algorithm, which gives the user the opportunity to provide the measured ILS as an input parameter, an option chosen for this study (Hase et al., 2004). This code is in wide use and has been thoroughly tested in the past for the HR125 as well as the EM27/SUN, e.g. Schneider and Hase (2009); Sepúlveda et al. (2012); Kiel et al. (2016a); Chen et al. (2016). Due to the low resolution of the



EM27/SUN, the atmospheric spectra were fitted by scaling of a priori trace gas profiles, although PROFFIT has the ability to perform a full profile retrieval (Dohe, 2013). As source of the a priori profiles, the TCCON daily profiles introduced in section 2.1 are utilized to be consistent with the TCCON analysis. Also for the daily temperature and pressure profiles, the approach from TCCON was adopted, using NCEP model data together with on site ground pressure data from a meteorological tall tower (www.imk.kit.edu/messmast/).

For the evaluation of the $O_2$ column the 7765 - 8005 $cm^{-1}$ spectral region is used, which is also applied in the TCCON analysis (Wunch et al., 2010). For $CO_2$ we combine the two spectral windows used by TCCON to one larger window ranging from 6173 to 6390 $cm^{-1}$. $CH_4$ is evaluated in the 5897 - 6145 $cm^{-1}$ spectral domain. For $H_2O$ the 8353 - 8463 $cm^{-1}$ region is used. This differs from TCCON, which deploys several narrow spectral windows, a strategy which is more in line with high resolution spectral observations. For consistency reasons, and to reference the results to the WMO scale, the EM27/SUN retrieval also performs a post-processing. The AICFs from TCCON are adopted, and similar to Wunch et al. (2010), an airmass dependency correction is performed, although other numerical values for the correction parameters are used. Details can be found in Frey et al. (2015); Klappenbach et al. (2015).

## 3 Long term performance

### 3.1 ILS analysis

Accurate knowledge of the real ILS of a spectrometer is extremely important because errors in the ILS lead to systematic errors in the trace gas retrieval. For this reason regular ILS measurements were performed from the beginning of this study four years ago to detect possible misalignments and alignment drifts. Source of a de-adjustment is mostly mechanical shock, due to e.g. impacts or vibrations especially due to transportation of the instruments. A detailed description of the ILS analysis is given in Frey et al. (2015). The time series of the ILS measurements is shown in Fig. 1, the modulation efficiency (ME) at maximum optical path difference (MOPD) ranges between 0.9835 and 0.9896, with a mean value of 0.9862 and a standard deviation of 0.0015. The phase error is close to zero for the whole time series with a mean value of 0.0019 ± 0.0018. This modulation efficiency is significantly different from nominal, which is surprising, as great care was taken to align the instrument. Therefore open path measurements were also performed for the HR125 at a resolution of 0.5 $cm^{-1}$ to investigate whether this method shows a bias. For this small optical path difference, the alignment of the HR125 should be very close to nominal. However, the LINEFIT analysis shows a ME of 0.9824 at MOPD. From this result it is concluded that this method shows an overall low bias of around 1.5 - 2 %, probably due to a slight underestimate of the pressure broadening parameters of $H_2O$ in the selected spectral region.

There is no overall trend apparent in the time series, the remaining differences in the modulation efficiency are probably due to the remaining uncertainty of the measurement technique. As is indicated by the more frequent measurements in 2017, there is also no seasonality in the results of the open path measurements. It should be noted that the measurement routine was refined in the course of this work. In particular, in the beginning (2014) it was assumed that the inside of the EM27/SUN is free of water vapor, so the instrument was not vented during the lamp measurements. However, sensitivity studies as presented in Frey et al.



(2015) revealed that the influence of the water vapor column inside of the spectrometer can not always be neglected. After this discovery the instrument was vented during the open path measurements. This is why the 2014 calculations show larger scatter, as here the amount of water vapor inside the spectrometer is not known. For this analysis it was assumed that also for the 2014 measurements the total pressure inside the spectrometer is the same as of the surrounding air, which is a sensible assumption

as the spectrometer is not evacuated. This explains also that the deviations become smaller in 2017. A further test to verify the stability of the instrument is the $X_{air}$ parameter, which is the surface pressure divided by the measured column of air. This test will be shown in section 3.3.

The grey lines in Fig. 1 denote transportation of the spectrometer over longer distances for field campaigns in Berlin (North-Eastern Germany), Oldenburg (Northern Germany) and Paris (France) and for maintenance at Bruker Optics. Note that no

realignment of the interferometer was performed during this maintenance. Only the reference HeNe laser was exchanged due to sampling instabilities during interferogram recordings. More specifically, the laser wavelength was unstable resulting in a corruption of parts of the measured spectra. Later in 2016 and 2017 this instrument was not used for campaigns since it has been chosen as the reference EM27/SUN for comparison measurements next to the HR125 spectrometer in order to take measurements at Karlsruhe as continuously as possible. The instrument was not realigned during the whole comparison study.

An error estimation for the open path measurements is given in Table 1. For the temperature and pressure error, the stated accuracies of the data logger manufacturer were used. For the other potential error sources reasonable estimates were made. The total error, given by the root-squares-sum of the individual errors, is 0.29 % in ME amplitude, consisting of several errors of approximately the same magnitude.

### 3.2 Total column time series

In this section the total column measurements from the EM27/SUN are compared to the reference HR125 spectrometer. For the measurements, the EM27/SUN was moved to a terrace on the top floor of the IMK-ASF, building 435 KIT CN (49.094 °N, 8.436 °E, 133 m a.s.l.), on a daily basis if weather conditions were favourable. The spectrometer was moved from the lab on the fourth floor to the roof terrace on the seventh floor thus being exposed to mechanical stress. The instrument was coarsely oriented north, without effort for levelling. If further orientation was needed, the spectrometer was manually rotated so that the

solar beam was centered onto the entrance window. The CamTracker program was then able to track the sun. The spectrometer was operated at ambient temperatures. During summer, the spectrometer heated up to temperatures above 40 °C. In order to protect the electronics from the heat, a sun cover for the EM27/SUN was built, which reduced the temperatures inside the spectrometer by about 10 °C. In winter the temperatures were as low as -4 °C at the start of measurements. Double-sided interferograms with 0.5 cm$^{-1}$ resolution were recorded. With 10 scans and a scanner velocity of 10 kHz, one measurement

takes about 58 s. For precise time recording, a GPS receiver was used.

The full time series from March 2014 to November 2017 is shown in Figure 2 for the three data sets. For better visibility only coincident data points measured within one minute between EM27/SUN and the other data sets are shown. There are 8349 paired measurements between EM27/SUN and TCCON and 4624 between EM27/SUN and HR125 LR, in total there are 50550 EM27/SUN and 25361 TCCON measurements.



All gases show a pronounced seasonal cycle, where the variability in water vapour is strongest with values below $1 \times 10^{26}$ molecules $m^{-2}$ in winter and up to $14 \times 10^{26}$ molecules $m^{-2}$ in summer. Furthermore, the seasonal cycle of water vapour is shifted with respect to the other species. Another feature seen is that there is an offset in the EM27/SUN (red squares) and HR125 LR (blue squares) total column data with respect to the TCCON data (black squares). The occurence of a systematic

bias when reducing the spectral resolution has been observed by several investigators (Petri et al., 2012; Gisi et al., 2012). The observed offset between EM27/SUN and HR125 LR measurements is smaller. The remaining difference can be attributed to the different measurement heights of the HR125 (112 m) and EM27/SUN (133 m). For a quantitative analysis we do not utilize the total column measurements but rather use the $X_{Gas}$, as in this representation systematic errors, e.g. ILS errors, timing errors, tracking errors and nonlinearities mostly cancel out. Furthermore the height dependence largely cancels out in

this representation. The comparison will be presented in the following sections.

Before, a sensitivity study is provided demonstrating the effect of changes in the ILS on the gas retrieval. For this one hour of measurements around solar noon on 01 August 2016 and 15 February 2017, corresponding to solar elevation angles (SEA) of 60° and 30°, were analysed with artificially altered ILS values. The results are shown in Table 2. An increase of 1 % in the modulation efficiency leads to a decrease of 0.35 % (0.37 %) on the retrieved $O_2$ column, 0.31 % (0.31 %) on $H_2O$, 0.26 %

(0.28 %) on $CH_4$ and 0.50 % (0.57 %) on $CO_2$ for an SEA of 60° (30°). So the change in the retrieved total column is not alike, but a unique characteristic of each species, and also slightly airmass dependent. As the decrease in the $CO_2$ column is larger than the decrease in the $O_2$ column, $XCO_2$ decreases with an increasing ME, 0.16 % (0.19 %) for 1 % ILS increase, whereas $XCH_4$ increases 0.10 % (0.09 %). This is opposed to prior studies (Gisi et al., 2012; Hedelius et al., 2016), reporting an increase of $XCO_2$ and decrease of $XCH_4$ for an increase of the modulation efficiency, albeit in agreement with the findings

from Hase et al. (2013) for the HR125 spectrometer, reporting that a change in the modulation efficiency results in a larger relative decrease in the $CO_2$ column than in the $O_2$ column.

## 3.3  $X_{air}$

In this section the column averaged amount of dry air ($X_{air}$) is investigated. This quantity is a sensitive test of the stability of a spectrometer because for $X_{air}$ there is no compensation of possible instrumental problems, in contrast to the DMFs, where

errors can partially cancel out. $X_{air}$ compares the measured oxygen column ($VC_{O_2}$) with surface pressure measurements ($P_S$):

$$X_{air} = \frac{0.2095}{VC_{O_2} \cdot \overline{\mu}} \cdot \left( \frac{P_S}{g} - VC_{H_2O} \cdot \mu_{H_2O} \right) \tag{1}$$

Here $\overline{\mu}$ and $\mu_{H_2O}$ denote the molecular masses of dry air and water vapour, respectively, $g$ is the column averaged gravitational acceleration and $VC_{H_2O}$ is the total column of water vapour. The correction with $VC_{H_2O}$ is necessary as the surface pressure instruments measure the pressure of the total air column, including water vapour. For an ideal measurement and

retrieval with accurate $O_2$ and $H_2O$ spectroscopy, as well as accurate surface pressure, $X_{air}$ would be 1. However, due to insufficiencies in the oxygen spectroscopy, this value is not obtained. For TCCON measurements $X_{air}$ is typically $\sim 0.98$ (Wunch et al., 2015). For the EM27/SUN prior studies showed a factor of $\sim 0.97$ (Frey et al., 2015; Hase et al., 2015; Klappen-





bach et al., 2015). Large deviations ($\sim 1$ %) from these values indicate severe problems, e.g. errors with the surface pressure, pointing errors, timing errors or changes in the optical alignment of the instrument. As mentioned in section 3.1, here $X_{air}$ is used to check whether the small changes in the modulation efficiency indicated by the open path measurements are due to actual alterations in the alignment of the EM27/SUN or due to the residual uncertainty of the calibration method.

The left panel of Figure 3 shows the $X_{air}$ time series of TCCON, the EM27/SUN and HR125 LR. For clarity, only conincident data points that were measured within one minute between the different data sets are shown. Grey areas denote periods where the EM27/SUN was moved over long distances for campaigns or maintenance. The absolute values of $X_{air}$ differ for the data sets, with $0.9805 \pm 0.0012$ for TCCON, $0.9669 \pm 0.0010$ for the EM27/SUN and $0.9670 \pm 0.0011$ for HR125 LR. The difference between the EM27/SUN and the HR125 LR is within $1\sigma$ precision. The difference between the EM27/SUN

and TCCON data set, which is commonly observed as previously noted, is a consequence of the different resolution together with the different retrieval algorithm (Gisi et al., 2012). It can be seen that all data sets exhibit a seasonal variability, which is more prominent in the TCCON data as can also be seen from the higher standard deviation. From this higher variability it can be concluded that the airmass dependency in the official TCCON $O_2$ retrieval is higher than for the PROFFIT retrieval, a finding also observed by Gisi et al. (2012). For the PROFFIT retrieval, it is suspected that part of the variability stems from

insufficiencies in the utilized HITRAN 2008 $H_2O$ linelist. It was reported by Tallis et al. (2011) that in the 8000-9200 $\mathrm{cm}^{-1}$ region, line intensities are low by up to 20 % compared to other wavenumber regions. This in return will lead to a systematic overestimation of the water column, which also affects $X_{air}$. To test the sensitivity of $X_{air}$ with respect to the measured $H_2O$ column, in the right panel of Figure 3 the original EM27/SUN time series is compared to a data set where the $H_2O$ column is artificially reduced by 20 %. This approach is further justified by a study from the Romanian National Institute for Research

and Development in Optoelectronics (INOE) conducted in 2017, where they compared total column amounts of water vapor from an EM27/SUN and a radiometer. They found that the EM27/SUN values were systematically higher by 20 % (Dragos Ene, priv. comm.). And indeed, the standard deviation, which is here used as a measure for the seasonal variability, of the modified time series (0.0009) is lower when compared to the original time series (0.0010).

There are no obvious steps between the EM27/SUN and the HR125 LR data sets so that it can be concluded that the EM27/SUN

is stable during the complete course of the over three year long comparison and differences seen in the modulation efficiency are introduced by the remaining uncertainty in the calibration method.

## 3.4   $XCO_2$

In Figure 4 $XCO_2$ time series of the three data sets are shown together with the offsets between the data sets. The general characteristics of the data sets are similar. The yearly increase of $XCO_2$ due to anthropogenic emissions of about 2 ppmv can

be seen as well as the seasonal cycle with a decrease of $XCO_2$ of approximately 10 ppmv during summer due to photosynthesis, characteristic for mid latitude stations. Despite these agreements in the general trend, there are also differences between the data sets. Relative to the TCCON data the EM27/SUN and the HR125 LR data sets are biased high (0.98 % and 0.84 % respectively). The scaling factors are calculated by taking the mean of all individual coincident point ratios (EM27/SUN / TCCON and EM27/SUN / HR125 LR), together with these ratios also a standard deviation is derived, see Table 3. A high





bias was also observed by Gisi et al. (2012); Frey et al. (2015), albeit with smaller absolute differences. This is due to the fact that (1) in the Gisi et al. paper the TCCON data was retrieved with an earlier version of GFIT (GGG2012) and (2) after the publication of the Frey et al. paper the Karlsruhe TCCON data was reprocessed with a customized GFIT retrieval accounting for baseline variations (Kiel et al., 2016b). The offset between EM27/SUN and TCCON shows a seasonal variability. Reasons

for this are mainly the differences in airmass correction, averaging kernels and retrieval algorithm. These effects have been investigated before (Gisi et al., 2012; Frey et al., 2015; Klappenbach et al., 2015; Hedelius et al., 2016, 2017). For the long term stability of the EM27/SUN the focus lies on the comparison with the HR125 LR data set, where the above mentioned differences cancel out. There is a small offset between the two data sets, resulting in a calibration factor of 1.0014, which is constant over time in the analyzed time period. To test this assumption a linear fit was applied to the $XCO_2$ ratios, see right

panel of Figure 4. In Table 3 the slope coefficient is depicted. For both comparisons the yearly trend in the ratio is well within the $1\sigma$ precision (0.44 ppmv) of the data set. In absolute numbers the slope per year is $\approx$ - 0.02 ppmv for both ratios, or a drift smaller than 0.1 ppmv over the whole comparison period of around three and a half years.

Figure 5 shows the data sets in a different representation. In the left panel the EM27/SUN is compared to the HR125 LR, the colorbar indicates the date of measurement and the dashed line is the 1 : 1 line. It can be seen that there is no trend in the

15 data apart from the overall increase in time due to anthropogenic emissions. In the right panel the EM27/SUN is compared to the TCCON data set, the colorbar shows the solar elevation angle (SEA). This representation is chosen so that the remaining airmass dependency of the ratio can be seen. It is also interesting to note that omitting the TCCON airmass independent correction factor (AICF) for our analysis would move the data set significantly closer to the 1 : 1 line. The scaling factor would change from 1.0098 to 0.9995. As this finding is not true for $XCH_4$ and is probably coincidental, we maintain the AICF.

**3.5 $XCH_4$**

Figure 6 shows the $XCH_4$ time series of the different data sets. As for $XCO_2$, the general features are in agreement for all data sets. There is a slight annual increase of about 10 ppbv. Also there is a seasonal cycle with a variability of $\approx$ 30 ppbv; however, compared to $XCO_2$ the interannual seasonality strength and phase varies significantly between the years due to the many different variable sinks and sources of methane, e.g. Dlugokencky et al. (1997). The differences between the data sets

largely resemble the differences observed for $XCO_2$. The bias between EM27/SUN and TCCON is 0.72 %, see Table 4. This bias is close to the bias observed by Hedelius et al. (2016), 0.75 %, where they used the GGG software package for the analysis of EM27/SUN spectra. Although a single bias is reported, as was observed for $XCO_2$ the offset is not constant, but rather shows a seasonality. The retrievals between EM27/SUN and HR125 LR agree within $1\sigma$ precision ($0.9997 \pm 0.0008$). The left panel of Figure 7 shows the ratio between EM27/SUN and HR125 LR color coded with the observation date. As for $XCO_2$,

no trend is apparent. An explicit linear fit to the $XCH_4$ ratio produces a slope coefficient of 0.0001, one order of magnitude smaller than the $1\sigma$ precision of the ratio (0.0008).

An interesting feature is observed in the ratio between EM27/SUN and TCCON data sets, see right panel of Figure 7. In general the pattern is similar to that of $XCO_2$, with a slight dependence on the SEA. The ratio in the figure is color coded





with the date of observation rather than the SEA. It can be seen that for 01 March 2016 and 14 March 2016 (shaded area in Figure 7) the $XCH_4$ ratio significantly differs from the other observations. Previous work by Ostler et al. (2014) has shown that stratospheric intrusion, caused for example by the subsidence of the polar vortex, has a different effect on MIR and NIR retrievals, even when using the same a priori profile. This is due to the differing sensitivity of the retrievals with respect to

altitude. Therefore, differences between the true atmospheric profile and the assumed a priori profiles on these days could cause the differences seen. This effect will also lead to larger differences between EM27/SUN and TCCON $XCH_4$ because of the different impact on the retrieved columns due to differing sensitivities. A spread of the polar vortex to mid-latitudes could lead to significantly altered $CH_4$ profiles compared to the a priori profiles, explaining the observed differences in the $XCH_4$ ratio.

The left panel of Figure 8 shows $N_2O$ data from the Microwave Limb Sounder (MLS) on the Aura satellite for several days in February and March 2016 on the 490 K potential temperature level, corresponding to a height of approximately 18 km. $N_2O$ is chosen because it serves as a tracer for the position of the polar vortex. Indeed it seems that beginning of March 2016 the polar vortex stretches out to mid-latitudes. To further test this hypothesis in the right panel of Figure 8 independent NDACC $CH_4$ profiles from the Jungfraujoch station in 2016 are shown. The station is situated approximately 270 km south of Karlsruhe with

a station height of 3580 m. For dates without measurements, the data was interpolated using a weighted average. The dotted black lines denote 1 March 2016 and 14 March 2016, the dates on which the $XCH_4$ ratio between EM27/SUN and TCCON shows an anomaly. The changed profile shape during that period is clearly visible. As this station is south of Karlsruhe, it is expected that also for Karlsruhe the $CH_4$ profile shows considerable downwelling, explaining the observed anomaly in the $XCH_4$ ratio.

## 20   4   Ensemble performance

Having investigated the long term stability of the EM27/SUN with respect to a reference spectrometer in the previous section, here the level of agreement of an ensemble of EM27/SUN spectrometers is presented. The procedure is the same as for the comparison between the reference EM27/SUN and the HR125. First, the ILS is analysed, followed by calibration factors for $XCO_2$ and $XCH_4$.

## 25   4.1   ILS measurements and instrumental examination

The measurement of the ILS is a valuable diagnostic for detecting misalignments of spectrometers. Differences in the ILS of the EM27/SUN spectrometers due to misalignment can lead to biases in the data products between the instruments. Here the spread of ILS values of all EM27/SUN spectrometers that were checked at KIT in the past four years is estimated. Numerical values are given in Table 5, the results are shown in Figure 9. The black square denotes an ILS measurement of the HR125

30     spectrometer, also with 1.8 cm MOPD. This test was done to check for an absolute offset of our method. The HR125 would be expected to show an ideal ILS for short optical path differences, but a value of 0.9824 was obtained. From this measurement it is concluded that our method shows an absolute offset and that values between 0.98 and 0.99 are desired.





In general, the agreement between the 30 tested EM27/SUN is good with an ensemble mean of $0.9851 \pm 0.0078$, which is not differing significantly from the value obtained for the HR125, but there are exceptions. Instrument SN 44 was checked at KIT only after an upgrade with the second channel was performed at Bruker Optics. Before realignment, the instrument showed a very low ME value of 0.9374. A realignment of the instrument enhanced the ME to 0.9714. This is still significantly low com-

5 pared to the EM27/SUN ensemble mean, but the difference was drastically reduced. The second instrument showing strong deviations from the ensemble mean is SN76 with an ILS of 1.0160, the only instrument showing overmodulation. The ILS was even higher (1.0350) when first ILS measurements were performed. Due to our findings, the manufacturer exchanged the beamsplitter which reduced the overmodulation, but it partly remained. In the meantime it was recognized as the cause of error that the manufacturer during assembling of the instrument forgot to insert the foreseen spacer to achieve the correct detector

position with respect to the beamsplitter. The beamsplitter is coated, and the coating is applied on both sides of the beamsplitter over half the surface area. If the optical axis of the detector element coinciding with the transition region of the two coating areas, detrimental effects occur. For this reason the detector element needs to be raised with respect to the interferometer. This problem occured for instrument SN 77 but there it was diagnosed and corrected by KIT (ILS before lifting: 1.0340, ILS after correction: 0.9855).

The above mentioned problems show the benefit of the calibration routine at KIT. Imperfections from nonideal alignments were diagnosed and corrected. Also other detrimental effects, e.g. double-passing, channeling, nonlinearity issues, solar tracker problems, inaccurate positioning of the second detector or camera issues, were corrected or minimized for a number of instruments. Finally, it was checked whether the linear interpolation method suppressing sampling ghosts was activated.

## 4.2 $XCO_2$ and $XCH_4$ comparison measurements

After checking the alignment and performing lamp measurements, side-by-side solar calibration measurements were performed on the terrace on top of the KIT-IMK office building with each spectrometer with respect to the reference EM27/SUN and also a co-located HR125 spectrometer. Calibration measurements started in June 2014 and are ongoing, if new spectrometers arrive for testing. The aim is to have at least one day of comparison measurements so that the spectrometers can be scaled to TCCON via the reference EM27/SUN. TCCON is extensively compared to measurements on the WMO scale. Dates of the comparison

measurements for the different spectrometers as well as number of coincident measurements are given in Table 6. On January 21 2016, our reference spectrometer suffered from laser sampling errors after approximately one hour of measurements. Therefore the number of coincident measurements for SN62 and 63 that were calibrated on this date are sparse. A typical calibration day is depicted in Fig. 10.

The calibration factors and standard deviations for all instruments with respect to the reference spectrometer are also depicted in Table 6. Calibration factors and standard deviations were obtained using the methods described in section 3.4. The calibration factors are close to nominal for all species and instruments. For $XCO_2$ the ensemble mean is high compared to the reference EM27/SUN, with a mean calibration factor of 0.9993. Applying this factor to all calculated calibration factors centers the data around the ensemble mean. As an estimate for the spread of the calibration factors $\frac{1}{n}\Sigma|X\text{Gas factor} - 1|$, we arrive at an





average bias between the instruments of 0.20 ppmv. From Table 6 we can also calculate an average standard deviation $\frac{1}{n}\Sigma|\sigma|$ of 0.13 ppmv. For $XCH_4$ the ensemble mean is closer to the reference EM27/SUN (0.9997) as compared to $XCO_2$. From this results an average bias of 0.8 ppbv. The average standard deviation is 0.6 ppbv. These values are comparable to results obtained in a study from Hedelius et al. (2017). They checked the intercomparability of the 4 United States TCCON sites using

an EM27/SUN as a traveling standard. They report average biases of 0.11 ppmv for $XCO_2$ and 1.2 ppbv for $XCH_4$, for the average standard deviations they obtain 0.34 ppmv ($XCO_2$) and 1.8 ppbv ($XCH_4$). It has to be noted that for the Hedelius et al. (2017) study only data within $\pm$ 2 h local noon was taken into account whereas here no constraints regarding the time of measurement were applied. As another sensitive test the $O_2$ total column calibration factors are given. In contrast to $XCO_2$ and $XCH_4$, there is no canceling of errors in this quantity. The ensemble mean is slightly high compared to the reference

EM27/SUN (0.9999). The average bias is 0.11 % $O_2$ with an average standard deviation of 0.04 % $O_2$.

Note that for our setup this average bias is a worst case scenario. The bias only applies if no calibration factor is used in the subsequent analysis. The strength of this calibration routine is that the computed calibration factors can be used, thereby significantly lowering the bias between different EM27/SUN spectrometers. The remaining bias is then given by the long term drift of the individual instrument, see section 3.4 and 3.5, and sudden alignment drifts due to mechanical strain from e.g.

transport, campaign use. To estimate this drift, we utilize the calibration factors before and after the Berlin campaign performed in 2014. There the drifts between five instruments were below 0.005 % $XCO_2$ and 0.035 % $XCH_4$ (Frey et al., 2015).

Ideally, we would expect identical calibration factors as we took the real ILS of the instruments into account. As this is not the case, we investigate whether the remaining differences can be attributed to the uncertainties of the open path measurements, which are summarized in Table 1. The results are incorporated in Fig. 11. The left panel shows the correlation between $O_2$ and

$XCO_2$ calibration factors. Black squares denote the empirical calibration factors derived from the side-by-side measurements. The red squares show calculated calibration factors based on the ME uncertainty budget. The dashed red line is a linear fit through the calculated factors. About half the measured empirical factors are within the bounds of the factors derived from the ME error budget. Furthermore the slopes of the calculated and empirical factors are in good agreement, confirming that the ME uncertainty is contributing to the uncertainty of the calibration factors. The other contributions for this uncertainty are due

to a superposition of various small device-specific imperfections. The right panel of Fig. 11 shows the correlation between $O_2$ and $XCH_4$ calibration factors. The findings mentioned above for the $O_2$ and $XCO_2$ correlation also hold true here.

## 5   Conclusions and Outlook

Based on a long-term intercomparison of column-averaged greenhouse gas abundances measured with an EM27/SUN FTIR spectrometer and with a co-located 125HR spectrometer, respectively, we conclude that the EM27/SUN offers highly stable

instrument characteristics on timescales of several years. The drifts on shorter timescales reported by Hedelius et al. (2016) were probably exclusively - as conjectured by the authors of the study - due to a deviation from the instrumental design as originally recommended. The application of a wideband detector suffering from nonlinearity together with steadily decreasing signal levels due to ageing of the tracker mirrors seem to be the reason for the observed drifts.



The favourable instrument stability which is preserved even during transport events and operation under ambient conditions suggests that the EM27/SUN spectrometer is well suited for campaign use and long-term deployment at very remote locations as a supplement of the TCCON. An annual to biannual check of the instrument performance by performing a side-by-side intercomparison with a TCCON spectrometer seems adequate for quality monitoring. For separating out instrumental drifts

from atmospheric signals, the addition of low-resolution spectra derived from the TCCON measurements is highly useful, because in this kind of comparison, the smoothing error and any possible resolution-dependent biases of the analysis software cancel out. The ensemble performance of 30 EM27/SUN spectrometers turns out to be very uniform, supported by a centralized acceptance inspection performed at KIT before the spectrometers are deployed. When using the empirical ILS parameters derived for each spectrometer, the scatter in $XCO_2$ amounts 0.13 ppmv, while it is 0.6 ppbv for $XCH_4$. The standard deviation

of the oxygen columns is 0.04%. We expect that the conformity of measurement results will be even better than indicated by this scatter, if the remaining empirical calibration factors are taken into account. These empirical calibration factors are likely composed of several small device-specific error contributions, a major contribution was identified to stem from the uncertainty of the ILS measurements.

Continuation and further development of the COCCON activities seem highly desirable for achieving the optimal performance

of the growing EM27/SUN spectrometer network. The implemented pre-deployment procedures of testing, optimizing, and calibrating each device - executed by experts at a central facilty – help to ensure consistent results from EM27/SUN spectrometers operated in any part of the world. This approach is corroborated by the proven excellent long-term stability of instrumental characteristics, and the proven high degree of stability under thermal and mechanical burdens as they occur during transport. The EM27/SUN spectrometer does not require continuing expert maintenance and it is very simple to operate, we therefore

expect that many investigators world-wide who are not keen to become FTIR experts will be attracted by this measurement device, operating it as a side activity. Current COCCON work supported by ESA in the framework of the COCCON PROCEEDS project will result in an easy-to-handle preprocessing tool optimized for the EM27/SUN spectrometer. This tool will generate quality-checked spectra from raw interferograms, which then are forwarded to a central data analysis facility. A demonstration setup of the central facility will be part of COCCON PROCEEDS. When finally implemented on an operational level, the

facility will remove the whole burden of the quantitative trace gas analysis from the operator and ensure the consistency of the trace gas analysis chain to the utmost degree. Furthermore it will enable a timely reanalysis of all submitted spectra after upgrades of the retrieval procedures and minimize the risk of data loss if operators for some reason are stopping their activity. Finally, this centralized facility will serve as a unique contact point for the data users.

*Competing interests.* The authors declare that they have no conflict of interest.

*Acknowledgements.* We acknowledge support by the ACROSS research infrastructure of the Helmholtz Association of German Research Centres (HGF).



We acknowledge support by the MOSES research infrastructure of the HGF.

We thank the National Center for Environmental Prediction (NCEP) for providing atmospheric temperature profiles.

We thank the NASA science team for providing MLS data from the Aura satellite.

We thank the Jungfraujoch NDACC team for providing Jungfraujoch FTIR data.

5   I. Morino and A. Hori contributed by procuring the NIES instrument and performing additional instrumental line shape measurements in Tsukuba.

We acknowledge funding from the Australian Space Research Program - Greenhouse Gas Monitoring Project, the Australian Research Council project DE140100178, and the Centre for Atmospheric Chemistry (CAC) Research Cluster supported by the University of Wollongong Faculty of Science, Medicine and Health.



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



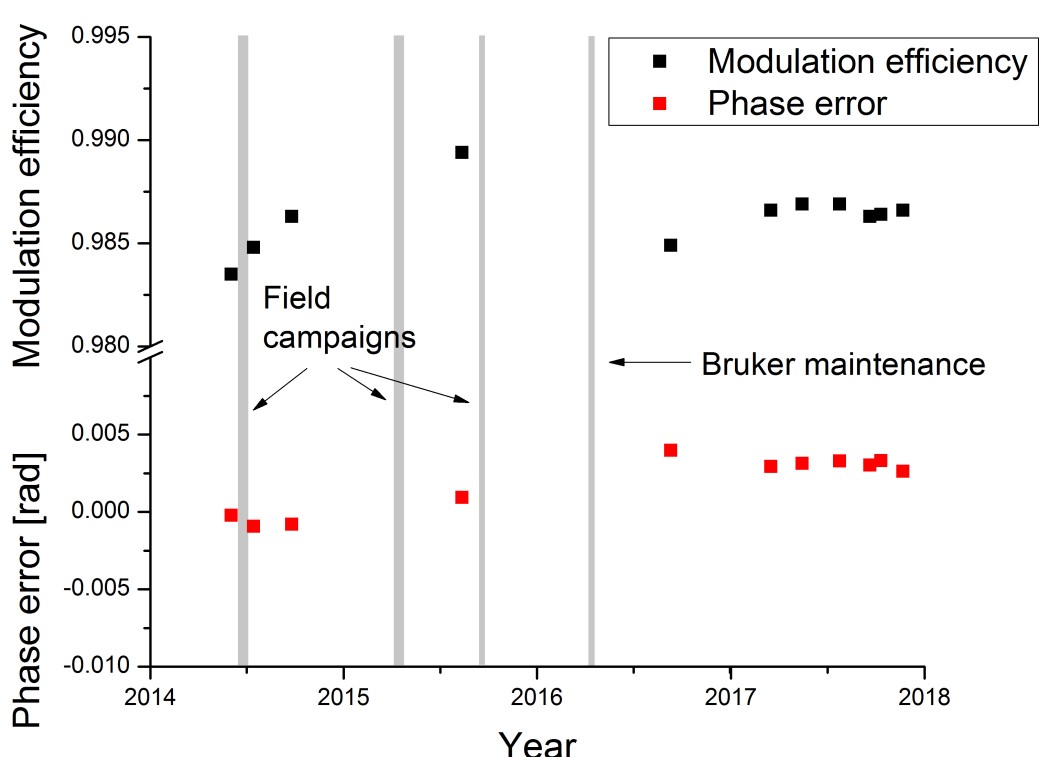

**Figure 1.** ILS time series of the reference EM27/SUN. Results for modulation efficiency and phase error were obtained with LINEFIT 14.5. The mean value of the modulation efficiency is 0.9862 with a standard deviation of 0.0015. For the phase error an average value of 0.0019 ± 0.0018 is observed. As can be seen from the closely spaced measurements in 2017, there is no seasonality in the ILS values. Grey areas denote periods of transportation of the instrument.





**Figure 2.** Total column time series for $O_2$, $CO_2$, $CH_4$ and $H_2O$ measured at KIT in Karlsruhe from March 2014 until October 2017. The number of interferograms and recording time for the different data types are the following: TCCON: 2 IFGs, 114 s; EM27/SUN: 10 IFGs, 58 s; HR125 LR: 4 IFGs, 152 s. Only coincident measurement points (within one minute) are depicted.





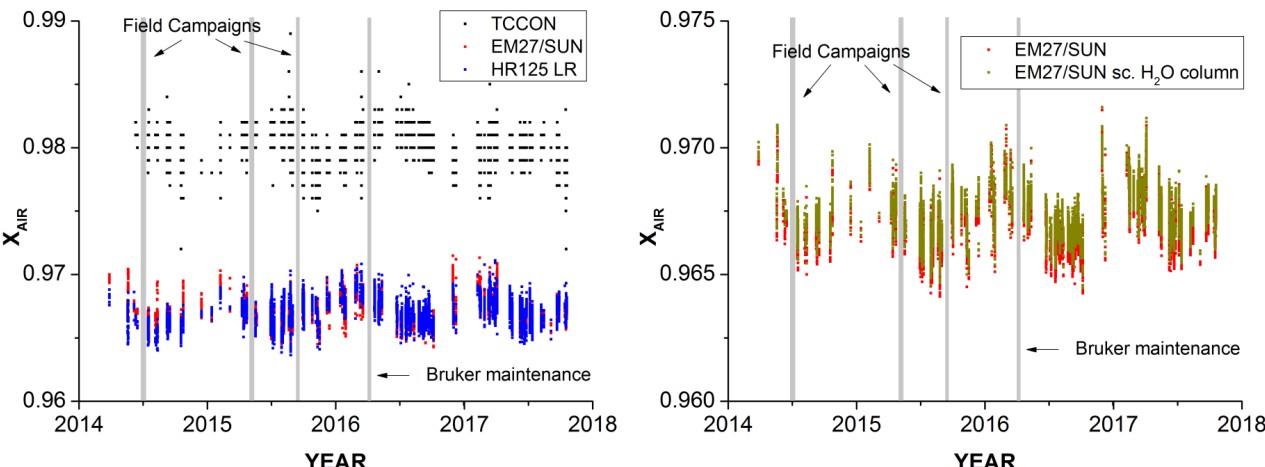

**Figure 3.** The left panel shows the $X_{air}$ time series measured at KIT in Karlsruhe for the TCCON, EM27/SUN and HR125 LR data sets. For clarity, only coincident measurements (within one minute) of the data sets are plotted. Grey areas denote periods where the EM27/SUN was moved over long distances. The right panel shows a comparison of the original EM27/SUN time series with a modified version, where a scaling factor of 0.8 was applied to the $H_2O$ total column.





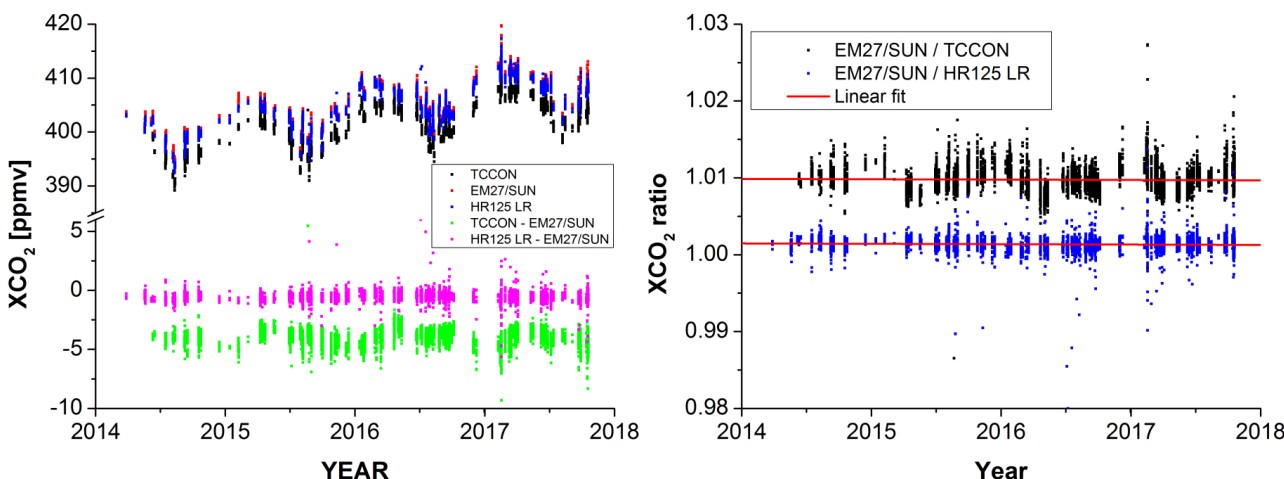

**Figure 4.** The left panel shows the $XCO_2$ time series measured at KIT in Karlsruhe for the three data sets from March 2014 to October 2017. Additionally the absolute offsets between the EM27/SUN and the two other data sets are shown. For clarity, only coincident measurements (within one minute) of the data sets are plotted. The right panel shows the $XCO_2$ ratio between the EM27/SUN and the two HR125 data sets. A linear fit was applied to investigate a possible trend in the ratios.





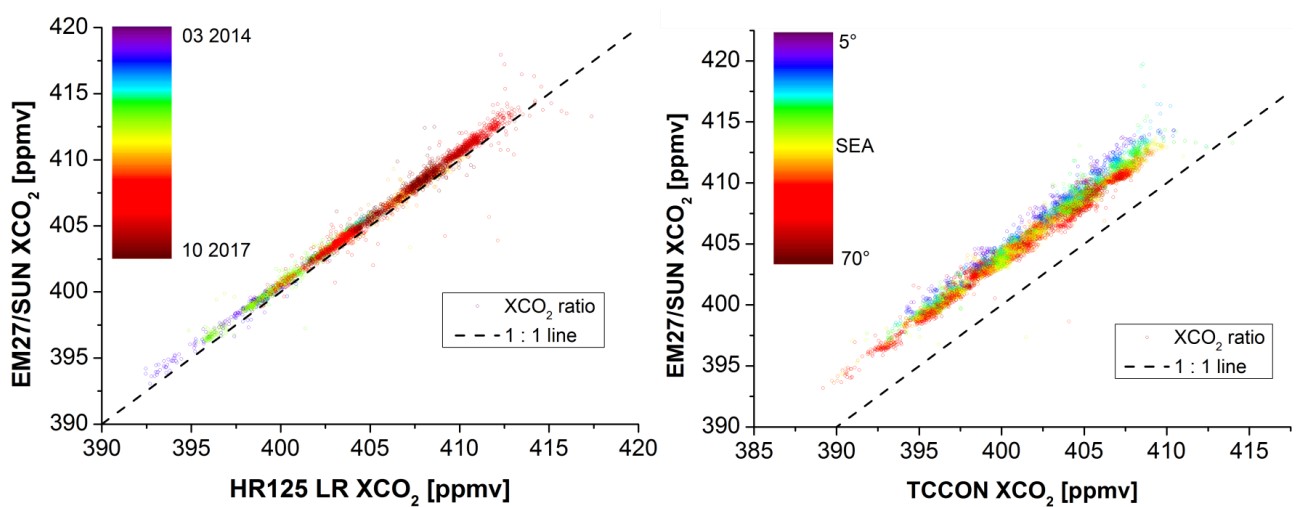

**Figure 5.** The left panel shows the $XCO_2$ comparison between EM27/SUN and HR125 LR. The colorbar denotes the date of the measurement, the dashed line is the 1 : 1 line. In the right panel the comparison with TCCON is shown. Note that here the colorbar shows the solar elevation angle.



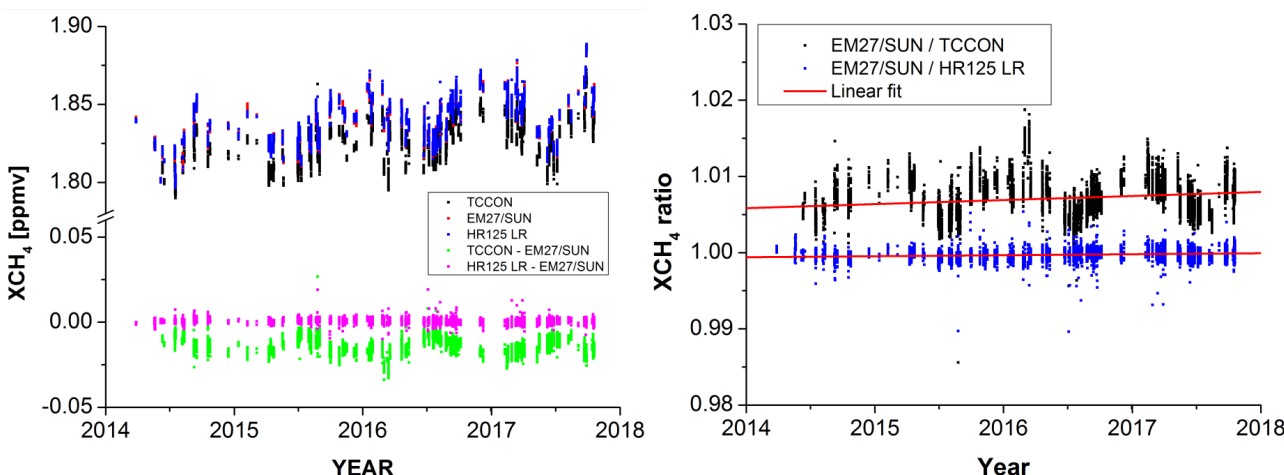

**Figure 6.** The left panel shows the $XCH_4$ time series measured at KIT in Karlsruhe for the three data sets from March 2014 to October 2017. Additionally the absolute offsets between the EM27/SUN and the two other data sets are shown. For clarity, only coincident measurements (within one minute) of the data sets are plotted. The right panel shows the $XCH_4$ ratio between the EM27/SUN and the two HR125 data sets. A linear fit was applied to investigate a possible trend in the ratios.



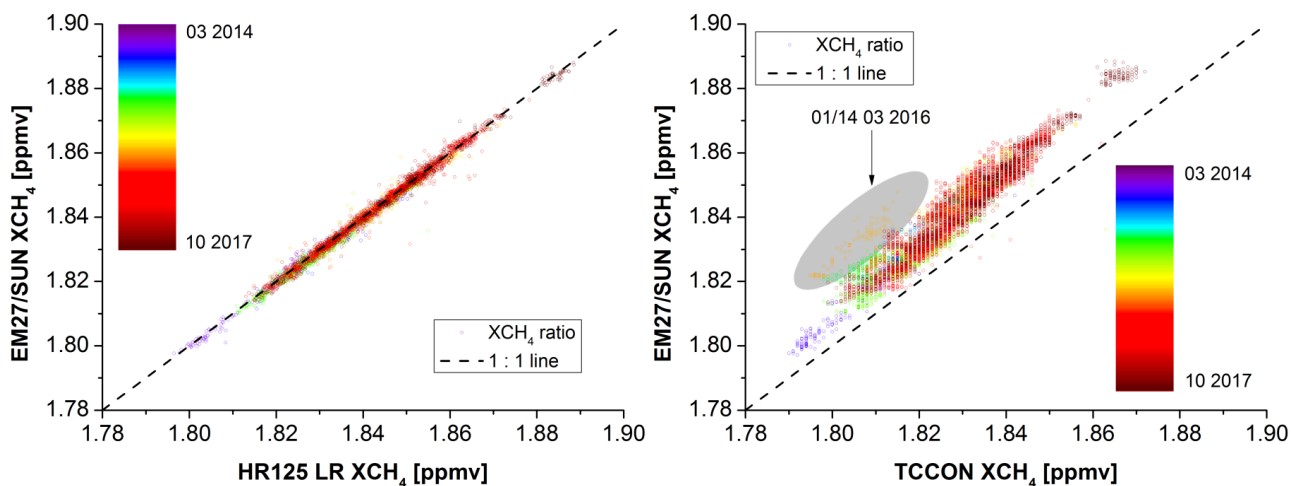

**Figure 7.** The left panel shows the $XCH_4$ comparison between EM27/SUN and HR125 LR. The colorbar denotes the date of the measurement, the dashed line is the 1 : 1 line. In the right panel the comparison with TCCON is shown. The shaded area encloses measurements from 01 and 14 March 2016. For these days the ratio is significantly different with respect to the remaining data set (see text for discussion).





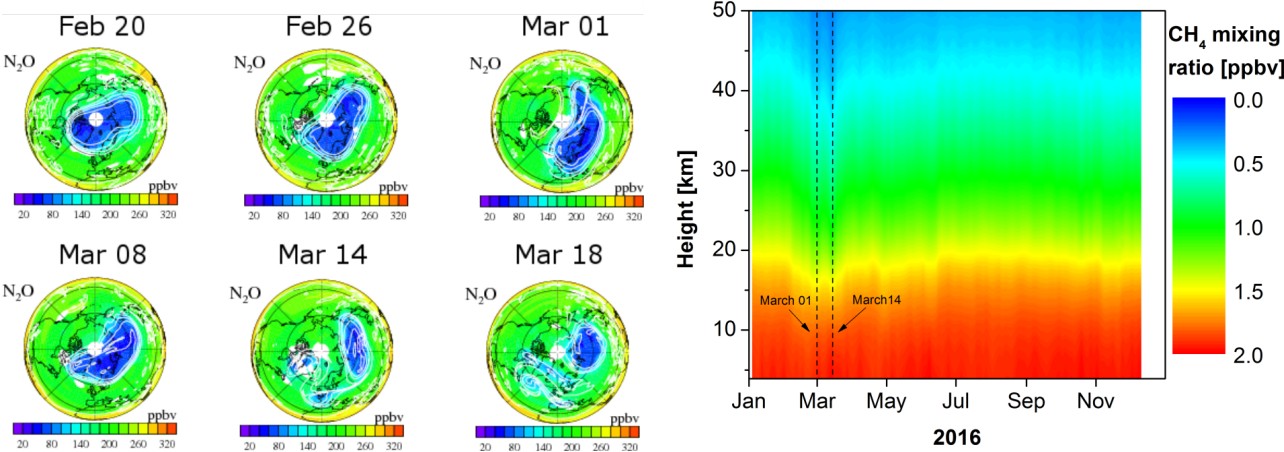

**Figure 8.** In the left panel $N_2O$ MLS data from the Aura satellite is shown as a tracer for the position of the polar vortex for several days in February and March 2016. Data and plots courtesy of the NASA science team (https://mls.jpl.nasa.gov/). The right panel shows $CH_4$ mixing ratios from the NDACC FTIR station Jungfraujoch in Switzerland, downloaded from the NDACC archive (http://www.ndaccdemo.org/stations/jungfraujoch-switzerland/). For dates with no measurements the data has been interpolated using a weighted average. Dotted lines depict March 01 and 14 2016. For these dates, the $XCH_4$ data significantly differs from the remaining data set.





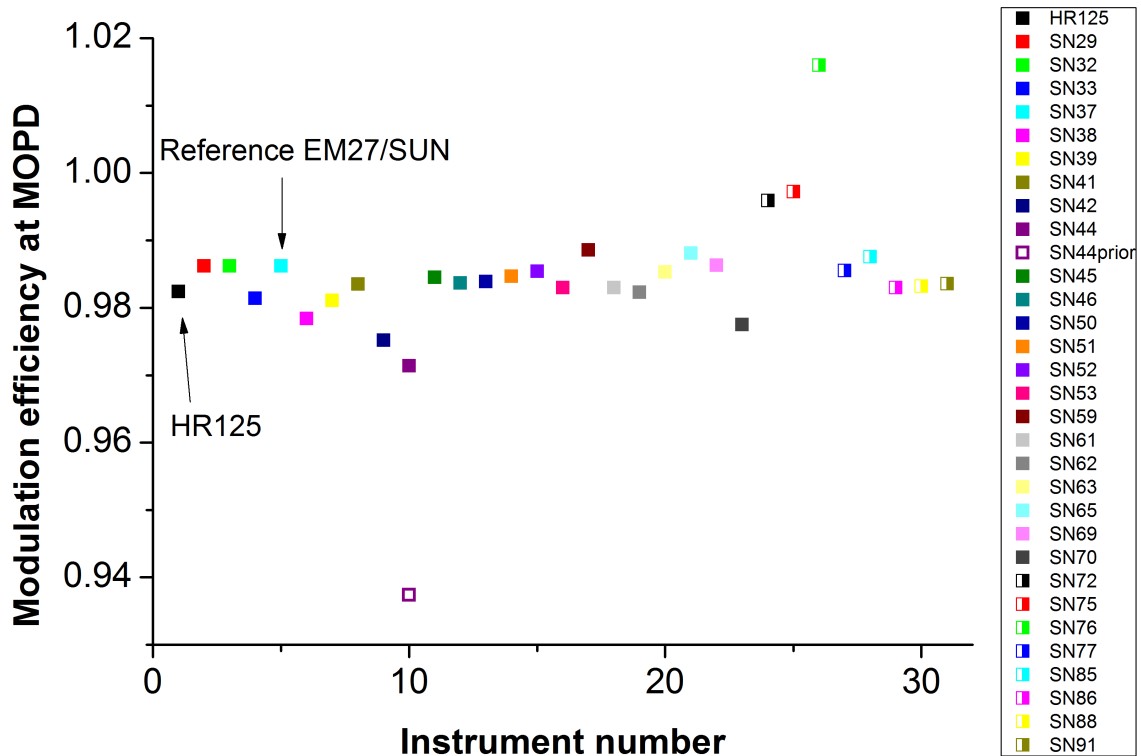

**Figure 9.** Modulation efficiencies at MOPD for all EM27/SUN calibrated in Karlsruhe. For SN44 prior, ILS measurements were taken before an alignment check and subsequent realignment of the instrument. For comparison reasons, also an ILS measurement for the HR125 was performed.



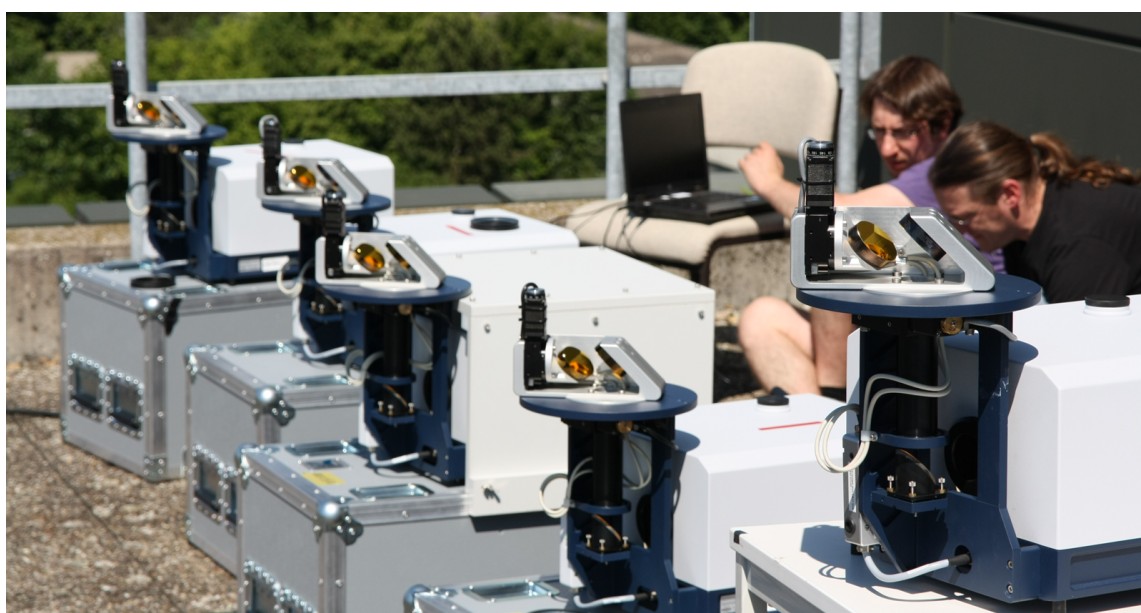

**Figure 10.** Calibration measurements performed on April 14 2015 on top of the KIT-IMK office building north of Karlsruhe.





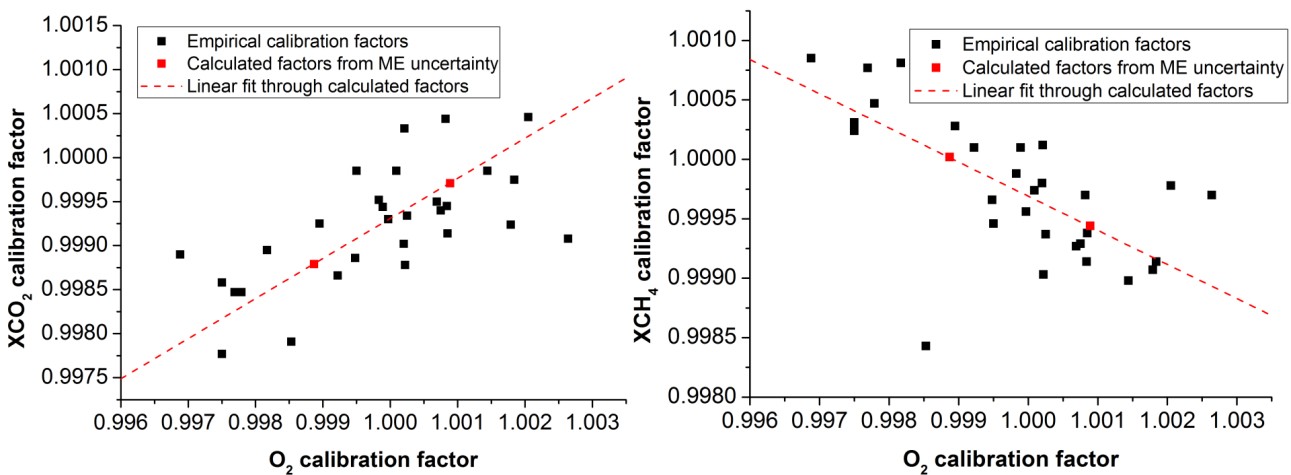

**Figure 11.** Correlation of $O_2$ calibration factors and $XCO_2$ (left panel) as well as $XCH_4$ (right panel) calibration factors. Black squares show the empirical calibration factors from the side-by-side measurements, red squares show calculated factors derived from the total ME uncertainty shown in Table 1, the dashed red line is a linear fit through the calculated factors. The slope of empirical and calculated factors is in good agreement.





**Table 1.** Estimated ME uncertainties for various error sources.

| Error source | Uncertainty | Propagation on ME |
|---|---|---|
| Temperature | $\pm$0.8 K | -0.16 % |
| Total pressure | $\pm$3 mbar | 0.19 % |
| Distance | $\pm$5 cm | -0.04 % |
| Partial pressure $H_2O$ | $\pm$0.5 mbar | 0.13 % |
| Measurement noise | | 0.05 % |
| Total | | 0.29 % |



**Table 2.** Sensitivity study on the effect of ILS changes on the retrieval of the total gas columns. Depicted is hourly pooled data on 01 August 2016 and 15 February 2017 around solar noon, corresponding to a solar elevation angle of 60° and 30°.

| ME | $O_2$ [$10^{28}$ molc m$^{-2}$] | $H_2O$ [$10^{26}$ molc m$^{-2}$] | $CH_4$ [$10^{23}$ molc m$^{-2}$] | $CO_2$ [$10^{25}$ molc m$^{-2}$] |
|---|---|---|---|---|
| August 2016 | | | | |
| 0.99 | 4.6097 | 7.4551 | 3.9457 | 8.7321 |
| 1.00 | 4.5936 | 7.4323 | 3.9356 | 8.6879 |
| February 2017 | | | | |
| 0.99 | 4.6718 | 3.7746 | 4.0261 | 9.0968 |
| 1.00 | 4.6545 | 3.7628 | 4.0148 | 9.0455 |



**Table 3.** $XCO_2$ biases between EM27/SUN and HR125 data sets.

| $XCO_2$ ratio | No. coincidences | Mean ( $1\sigma$) | Yearly trend in the ratio |
|---|---|---|---|
| EM27 / TCCON | 8349 | 1.0098 (0.0015) | $-5 \times 10^{-5}$ |
| EM27 / HR125 LR | 4624 | 1.0014 (0.0011) | $-5 \times 10^{-5}$ |





**Table 4.** XCH$_4$ biases between EM27/SUN and HR125 data sets.

| XCH$_4$ ratio | No. coincidences | Mean ( 1$\sigma$) | Yearly trend in the ratio |
|---|---|---|---|
| EM27 / TCCON | 8349 | 1.0072 (0.0024) | 0.0005 |
| EM27 / HR125 LR | 4624 | 0.9997 (0.0008) | 0.0001 |



**Table 5.** Summary of the modulation efficiencies at MOPD and phase errors for all EM27/SUN calibrated in Karlsruhe.

| Instrument SN | ME at MOPD | Phase error [rad] |
| --- | --- | --- |
| 29 | 0.9862 | 0.0014 |
| 32 | 0.9862 | 0.0034 |
| 33 | 0.9814 | -0.0017 |
| 37 | 0.9862 | 0.0019 |
| 38 | 0.9784 | 0.0009 |
| 39 | 0.9811 | -0.0005 |
| 41 | 0.9835 | 0.0001 |
| 42 | 0.9752 | 0.0039 |
| 44 | 0.9714 | -0.0019 |
| 44 (prior) | 0.9374 | -0.0074 |
| 45 | 0.9845 | 0.0034 |
| 46 | 0.9837 | 0.0024 |
| 50 | 0.9839 | 0.0023 |
| 51 | 0.9847 | 0.0017 |
| 52 | 0.9854 | 0.0048 |
| 53 | 0.9830 | 0.0025 |
| 59 | 0.9886 | 0.0029 |
| 61 | 0.9830 | 0.0013 |
| 62 | 0.9823 | 0.0053 |
| 63 | 0.9853 | 0.0011 |
| 65 | 0.9881 | 0.0024 |
| 69 | 0.9863 | 0.0030 |
| 70 | 0.9775 | 0.0056 |
| 72 | 0.9959 | 0.0030 |
| 75 | 0.9972 | 0.0041 |
| 76 | 1.0160 | 0.0007 |
| 77 | 0.9855 | 0.0016 |
| 85 | 0.9876 | 0.0025 |
| 86 | 0.9830 | 0.0031 |
| 88 | 0.9832 | 0.0007 |
| 91 | 0.9836 | 0.0021 |


**Table 6.** Calibration factors for $XCO_2$, $XCH_4$ and $O_2$ for all investigated instruments with respect to the reference EM27/SUN spectrometer as well as calibration dates and number of coincident measurements. Values in brackets denote percent standard deviations.

| Instr. SN | Dates | No. co. | $XCO_2$ factor | $XCH_4$ factor | $O_2$ factor |
|---|---|---|---|---|---|
| 29 | 140606, 140718 | 490 | 1.0004 (0.02) | 0.9997 (0.03) | 1.0008 (0.03) |
| 32 | 150414 - 150422 | 1548 | 0.9997 (0.03) | 0.9997 (0.03) | 1.0004 (0.03) |
| 33 | 170807, 170815 | 339 | 0.9991 (0.03) | 0.9994 (0.04) | 1.0009 (0.05) |
| 38 | 150410 - 150421, 160121 | 1609 | 0.9989 (0.03) | 0.9997 (0.04) | 0.9988 (0.04) |
| 39 | 140717, 150414, 150415 | 1210 | 0.9992 (0.04) | 0.9994 (0.04) | 1.0003 (0.04) |
| 41 | 140717, 150414 - 150422 | 1877 | 0.9999 (0.03) | 1.0002 (0.03) | 0.9991 (0.03) |
| 42 | 160730, 160801 | 368 | 0.9978 (0.04) | 1.0003 (0.04) | 0.9975 (0.03) |
| 44 | 170227 | 286 | 0.9979 (0.03) | 0.9984 (0.03) | 0.9985 (0.03) |
| 45 | 170807, 170815 | 382 | 0.9995 (0.03) | 0.9991 (0.04) | 1.0008 (0.02) |
| 46 | 170808, 170815 | 503 | 0.9993 (0.03) | 0.9994 (0.03) | 1.0003 (0.03) |
| 50 | 150421, 150422 | 699 | 0.9999 (0.03) | 0.9995 (0.03) | 0.9995 (0.03) |
| 51 | 160126, 160129 | 256 | 0.9995 (0.03) | 0.9993 (0.03) | 1.0007 (0.05) |
| 52 | 150421, 150422 | 727 | 0.9990 (0.04) | 0.9998 (0.05) | 1.0002 (0.05) |
| 53 | 150421, 150422 | 729 | 0.9987 (0.03) | 1.0001 (0.03) | 0.9992 (0.04) |
| 59 | 160318 | 273 | 0.9998 (0.03) | 0.9991 (0.03) | 1.0019 (0.04) |
| 61 | 151002, 170713 | 618 | 0.9993 (0.03) | 0.9996 (0.04) | 1.0000 (0.04) |
| 62 | 160121 | 18 | 0.9988 (0.04) | 0.9990 (0.02) | 1.0002 (0.02) |
| 63 | 160121 | 15 | 1.0003 (0.05) | 1.0001 (0.05) | 1.0002 (0.07) |
| 65 | 160511 | 234 | 1.0005 (0.04) | 0.9998 (0.05) | 1.0020 (0.03) |
| 69 | 160908, 170713 | 636 | 0.9994 (0.03) | 0.9993 (0.03) | 1.0008 (0.03) |
| 70 | 160831, 160906 | 522 | 0.9985 (0.02) | 1.0005 (0.03) | 0.9978 (0.03) |
| 72 | 170215, 170216 | 433 | 0.9994 (0.05) | 1.0001 (0.03) | 0.9999 (0.04) |
| 75 | 170516, 170517 | 852 | 0.9993 (0.03) | 0.9991 (0.03) | 1.0018 (0.05) |
| 76 | 170608 | 365 | 0.9991 (0.04) | 0.9997 (0.04) | 1.0026 (0.06) |
| 77 | 170927 | 389 | 0.9999 (0.03) | 0.9997 (0.03) | 1.0001 (0.04) |
| 85 | 180213, 180214 | 371 | 0.9993 (0.03) | 1.0003 (0.03) | 0.9990 (0.03) |
| 86 | 180213, 180214 | 464 | 0.9986 (0.03) | 1.0002 (0.03) | 0.9975 (0.05) |
| 88 | 180314 | 154 | 0.9990 (0.03) | 1.0008 (0.03) | 0.9982 (0.03) |
| 91 | 180228 | 148 | 0.9985 (0.03) | 1.0008 (0.03) | 0.9977 (0.04) |