# Peer review of "Building the COllaborative Carbon Column Observing Network (COCCON): Long term stability and ensemble performance of the EM27/SUN Fourier transform spectrometer"

_Atmospheric Measurement Techniques, 2018_

## Referee Comment (RC1) · Anonymous Referee #1 · 8 Jul 2018

The manuscript "Building the COllaborative Carbon Column Observing Network (COC-CON): Long term stability and ensemble performance of the EM27/SUN Fourier transform spectrometer" by Frey et al. assesses the stability of Bruker EM27/SUN spectrometers and evaluates their use for greenhouse gas (GHG) observations. EM27/SUN spectrometers are the spectrometers used in the recently founded COllaborative Carbon Column Observing Network (COCCON). The EM27/SUN spectrometers are portable and easy to operate. They have been used to quantify GHG emission sources (cities, exploration sites, ...) by column budgeting and are also used at permanent sites

to complement the well-established Total Carbon Column Observing Network (TC-CON). Therefore, COCCON will increase the number of sites, which perform groundbased column observations of GHGs. This is of high importance for the validation of GHG satellite retrievals. Since a high stability and small bias are vital for a network of GHG observations, this paper is scientifically of very high interest. The manuscript is well written and in my opinion it should be published after minor revisions.

Below are several point that should be addressed prior acceptance.

Abstract: The abstract should be more quantitative and clearer. All the information I am requesting is contained somewhere in the manuscript but in my opinion it should be mentioned in the abstract, e.g. "the EM27/SUN is stable on timescales of several years" How stable, how many years? "average bias across the ensemble" Is the average a good measure? What about min/max? "the application of these empirical factors is expected to further improve" The abstract should contain a number for the bias after all corrections and one before. Future papers will cite this number and quite often this is only taken from the abstract and not from the text.

p.2, line 11: "Furthermore these measurements can be directly used to evaluate emissions reductions as demanded by international treaties..." I do not know a study, where such measurements have been used to evaluate emission reductions. Please give a reference or mention that this is a future plan.

p.6, 3.2 It is mentioned that the EM27/SUN spectrometers are operated at a significantly different temperatures. It would be interesting to have a separate assessment of the temperature on the EM27 retrievals.

p.8, line 32: Why do HR125 LR and EM27/SUN have a different bias compared to TCCON?

p.9, line 4: "The offset between EM27/SUN and TCCON shows a seasonal variability. Reasons for this are mainly the differences in airmass correction, averaging kernels

and retrieval algorithm." Maybe this is discussed in the papers mentioned. However, it is highly important for the network. Therefore the reasons should be (re-) discussed here.

p.11/12, 4.2: The instrument dependent calibration factors: It would be good to elaborate the discussion about them and have the numbers summarised in a table, e.g. overall biases a) with instrument dependent cal factor (including uncertainties on the cal factor) and b) w/o instrument dependent cal factor.

p.13 line 9: define how you quantify "scatter"

Figure 1: How much information is coming from the apriori in the ILS retrieval? If the constraints are high one can always get a stable ILS. Is it possible to add averaging kernels or similar for the ILS retrieval? At least some explanation should be included in section 3.1.

Figure 2: I am not sure if this figure is really needed.

Figure 4 and 6: The difference in the left figures is difficult to see and contains the same information than the ratio in the figures on the right. Therefore I would leave the difference out in the figures on the left.

Figure 10: I would delete this.

---

## Referee Comment (RC2) · Anonymous Referee #2 · 9 Jul 2018

Overall comments:

This paper analyzes multi-year analysis of EM27/SUN results compared to TCCON. The long-term performance and stability of the EM27/SUN systems is important to use EM27/SUN results for science analysis and satellite validation. The EM27/SUN systems have potential as lower cost stationary instruments, and for use in shorter term field campaigns since EM27/SUN are easier to move.

I agree with reviewer 1 that column averaging kernels should be shown and compared

to TCCON and LR TCCON.

The assessment of EM27/SUN results relies on comparisons to a specially processed, modified TCCON dataset, called LR TCCON. LR TCCON is reduced resolution TC-CON, with a differently derived ILS, and processed with the PROFFIT software. However, LR TCCON has not itself been validated.

Significant differences are seen between EM27/SUN and the full resolution TCCON (shown in Figures 4 and 6) for XCO2 and XCH4. These errors should be quantified in the paper. The errors are seaonally dependent and look to have peak-to-peak seasonal errors of about 1 ppm for XCO2 and 20 ppb for XCH4, larger than the TCCON errors compared to aircraft validation (0.4 ppm for XCO2 and 5 ppb for XCO2 for GGG2014 (Wunch, 2015)). Comparisons of EM27/SUN results to LR TCCON are very good. However, LR TCCON has NOT been validated and comparisons of EM27/SUN versus LR TCCON is NOT validation of the EM27/SUN results and does NOT tie EM27/SUN to WMO.

In summary, if LR TCCON can be validated versus aircraft/AirCore with similar errors as the standard TCCON, then this paper will set useful limits on EM27/SUN errors. As the paper stands, validation that must be considered is versus the standard TCCON product, which is marginal for satellite validation and on the high side for other uses.

**Specific comments**

Introduction:

The COCCON project should be introduced in the introduction, with the objectives of the COCCON, and who is participating in COCCON, the length of the project (for example).

In the introduction, add in the importance of TCCON for OCO-2 and GOSAT validation, adding a sentence after line 23 something like: "TCCON stations are also the primary validation for OCO-2 (cite https://oco.jpl.nasa.gov/files/ocov2/OCO-

**AMTD**
2\_SciValPlan\_111005\_ver1\_0\_revA\_final\_signed1.pdf) and validating the satellite observations at different locations is critical for the validation effort (Wunch et al., 2017)."

A figure showing the TCCON (original and degraded resolution) and EM27 spectral range and radiance would be helpful for the reader, or a reference to a previous paper showing this.

The spectral ranges and approximate resolution should be given in wavelength in addition to wavenumber. Some scientists are used to wavelength and the translation is not immediately obvious.

Section 2.2 The description of the HR125 low resolution data set should include the software used to analyze it. I infer it is PROFFIT, but should be stated.

Page 5, line 15. Define ILS, modulation efficiency, phase error.

Page 5, line 22. How is the phase error calculated– describe or cite a reference. Why is phase error important? What does it affect?

The statement on line 7, page 7, "The remaining difference can be attributed to the different measurement heights of the HR125 (112 m) and EM27/SUN (133 m)." This needs to be further explained and quantified. Is it the total column? It would be useful to the reader to have a calculation accounting for the offset.

Table 2, it would be useful to show the effect on XCO2, etc, which is the key result. The reader looks between columns and thinks it will probably cancel for XCO2 but is not sure.

Page 8, line 11, "From this higher variability it can be concluded that the airmass dependency in the official TCCON O2 retrieval is higher than for the PROFFIT retrieval, a finding also observed by Gisi et al. (2012)." This statement needs to be modified for clarity to "...higher than for the PROFFIT retrieval on reduced resolution TCCON measurements."

AMTD
Page 8, line 25. "There are no obvious steps between the EM27/SUN and the HR125 LR data sets so that it can be concluded that the EM27/SUN is stable." The offset versus time needs to be quantified as well. Step functions and slower drift are both important to quantify.

Page 9 line 7. The green line on Fig. 4 shows significant differences between TCCON and EM27, on the order of 1 ppm it looks like. This seasonal cycle amplitude difference should be quantified. The pink difference (comparison to LR TCCON to EM27) looks very good. As stated in the overall comments, if the difference of EM27/SUN vs. TCCON is larger than the reported TCCON error, then it is important to determine the cause of this difference. PROFFIT should be applied to the full resolution TCCON data, OR GFIT should be applied to the low resolution TCCON data to separate out the PROFFIT/GFIT differences vs. ILS/truncation differences to determine the source of the difference between full-resolution TCCON and LR TCCON.

LR TCCON needs to be validated versus aircraft/AirCore before it can be used to validate EM27/SUN.

Similar comment for XCH4. In Fig. 6, differences for XCH4 between EM27 and full resolution TCCON look to have seasonal differences of about 20 ppb, which is higher than the TCCON estimated XCH4 error of 5 ppb.

Wording/formatting suggestions:

Line 11, suggestion: change "as demanded by" to "as specified by"

Line 16, word suggestion: "Nonetheless" change to "However" Line 20: "However, recently OCO-2 data was used for estimating the source strength of power plants (Nassar et al., 2017)", would reword to emphasize coverage issues, "Recently OCO-2 data was used for estimating the source strength of power plants (Nassar et al., 2017). However, this can only be done for power plants that lie directly under the OCO-2 overpass locations."
Make the dots bigger on the Fig 2-7 legends. It is very hard to tell which dot is blue and which is black in the legend.

Page 7 line 11, "Before, a sensitivity study is provided demonstrating the effect of changes in the ILS on the gas retrieval." I think change "Before" to "First".

I see reviewer 1 suggests deleting Fig. 10. However I think Fig. 10 is useful to show the size of the instrument. Perhaps make this figure small.

---

## Short Comment (SC1) · 15 Jul 2018

The authors may want to consider these comments in preparation for their final submission of this paper.

Abstract: It may be helpful to emphasize that the instrument is solar-viewing.

P1L3 – The word "stable" is used throughout. The authors use the term to both refer to 1) mechanical stability of the instrument, and 2) comparability of the retrievals to another product. Because this is a subject term (e.g., one person may say 0.5% accuracy is stable, and another may say 0.05%) it would be useful if the author's metric of stability was defined numerically. In the future, requirements for "stability" may change as well.

P1L5 – It may be useful to list the QA measures here, as the authors use several.

P2L2 – "Very uniform" is also subjective. It would be helpful to mention indicators of uniformness here in case readers only see the abstract.

P2L13 - Numerically, what is the reference precision of the TCCON?

P2L14 – Not only do 125HR instruments require more frequent maintenance than EM27/SUN instruments, it also needs to be done on site.

P2L20 –Ye et al. (ACPD, 2017) https://doi.org/10.5194/acp-2017-1022 recently estimated city/urban emissions using satellite observations. Data from other current and future satellites may be used to estimate emissions from more localized sources, but that remains to be seen.

P2L25 – "Low-cost" is subjective, but I would actually say the EM27/SUN spectrometer is quite expensive, and cost-prohibitive for many institutions to own. The authors may consider stating the 2018 price range for these instruments.

P2L29 & throughout – The authors often use the world "calibrated" when "compared" or "scaled to" would be a better choice in this context. Calibrated is usually reserved for values more directly measured and compared to a standard.

P3L15 – Define IMECC

P3L24 - Define/describe the NCEP data

P4L4 – What does "nominal" mean here and throughout?

Sect. 3.1 - The ME at MOPD is consistently around 0.985, so what should users running PROFFIT use for the ILS? Should the input ME at MOPD be 1.0? What was

used in this study (e.g., what does "real ILS" mean on P12L17)?

P8L23 – Given that Dragos Ene is a coauthor of this study it seems strange to use "they" instead of "we." The authors may consider removing the private communications citation and instead put in an author contribution section at the end: see Manuscript composition -> 14. Author contribution under https://publications.copernicus.org/for\_authors/manuscript\_preparation.html

P9L7 – I agree with Reviewer #2 that the focus on comparing with a LR rather than an HR dataset from the 125 HR instruments is dissatisfying. I would expect the additional information in HR data should at least make it possible to construct a dataset with smaller absolute errors and biases. If 2 Xgas measurements have large, but equal errors or biases they will agree well.

P13L19 – I would disagree that no maintenance is ever required. In my experience at least 6 of 9 EM27/SUN instruments I have been on campaigns with required some form of maintenance within their first two years. Even the reference spectrometer in this study needed maintenance in 2016. However, an advantage is they do not need to be maintained on-site, but rather can be shipped back to Bruker or KIT.

P13L21 – From here and the TCCON meeting the COCCON PROCEEDS sounds like a very exciting upcoming development. I think this project deserves a more complete description earlier on in the paper. I also agree that a more concrete description of COCCON will be useful.

P13L29 – Perhaps the authors may want to check with the editor, but there may be some conflicts of interest that should be declared (https://www.atmospheric-measurement-techniques.net/about/competing\_interests\_policy.html). For example, receiving research funding from, or working for a commercial company could be considered a conflict of interest per the Copernicus policy.

Figure 4 – The authors may consider changing the y-axis scale. Scales of 15 ppm, and

5% ( $\sim$ 20 ppm) are, in my opinion, quite large and make it difficult to judge comparability of the retrievals on shorter timescales. Especially as the satellite community is pushing towards accuracy of 1 ppm ( $\sim$ 0.25%) or better for XCO2.

Metrics of stability in the Xgas retrievals in addition to the linear fit over the full time series may be useful in the text. For example, on different timescales such as months or seasons – especially since differences on these timescales are quite noticeable. This will help if the COCCON is used in satellite validation to know if comparisons should only be done over multi-annual scales to get an overall bias as high and low values will cancel out, or if shorter time-scales are plausible. Seasonal or month-to-month biases would also lead to artificial cycles in global assimilation models.

Table 1 – It would be helpful to have a caption as to why some uncertainties always propagate to negative on ME.

Table 5 - Would all the authors advise that regular ILS monitoring is unnecessary and other EM27/SUN operators just use the values in this Table?

Table 6 - Would the authors recommend instrument operators not make their own sideby-side comparison at the beginnings and ends of instrument campaigns, and instead use these scaling factors?

---

## Author Comment (AC1) · 4 Dec 2018

Dear anonymous referee 1, dear Prof. Aben,

please find our reply and a revised version of the manuscript with tracked changes attached.

We hope that we could adequately address the referees concerns and are looking forward to the finalisation of the review process.

Yours sincerely Matthias Frey

Please also note the supplement to this comment:
https://www.atmos-meas-tech-discuss.net/amt-2018-146/amt-2018-146-AC1-supplement.zip

---

## Author Response (AR2)

Authors' answer to the interactive comments of anonymous referee #1 on "Building the COllaborative Carbon Column Observing Network (COCCON): Long term stability and ensemble performance of the EM27/SUN Fourier transform spectrometer" by Frey et al., Atmos. Meas. Tech. Discuss., amt-2018-146

First of all, we would like to thank the anonymous referee #1 for the help in further improving the current presentment by a thorough assessment with regards of content and the careful technical proofreading resulting in the identification of several imprecisions and typos.

Referee: "The manuscript "Building the COllaborative Carbon Column Observing Network (COCCON): Long term stability and ensemble performance of the EM27/SUN Fourier transform spectrometer" by Frey et al. assesses the stability of Bruker EM27/SUN spectrometers and evaluates their use for greenhouse gas (GHG) observations. EM27/SUN spectrometers are the spectrometers used in the recently founded Collaborative Carbon Column Observing Network (COCCON). The EM27/SUN spectrometers are portable and easy to operate. They have been used to quantify GHG emission sources (cities, exploration sites, ...) by column budgeting and are also used at permanent sites to complement the well-established Total Carbon Column Observing Network (TCCON). Therefore, COCCON will increase the number of sites, which perform groundbased column observations of GHGs. This is of high importance for the validation of GHG satellite retrievals. Since a high stability and small bias are vital for a network of GHG observations, this paper is scientifically of very high interest. The manuscript is well written and in my opinion it should be published after minor revisions. Below are several point that should be addressed prior acceptance.

Abstract: The abstract should be more quantitative and clearer. All the information I am requesting is contained somewhere in the manuscript but in my opinion it should be mentioned in the abstract, e.g. "the EM27/SUN is stable on timescales of several years" How stable, how many years? "average bias across the ensemble" Is the average a good measure? What about min/max? "the application of these empirical factors is expected to further improve" The abstract should contain a number for the bias after all corrections and one before. Future papers will cite this number and quite often this is only taken from the abstract and not from the text."

**Authors**: As suggested by the referee, we underpin the qualitative statements of the previous version ("stable", "very uniform") with the quantitative results given in sections 3.4, 3.5, and 4.2 of the paper. In addition to the robust metric based on absolute differences, we calculate the standard deviation among the empirical calibration factors of all spectrometers (this metric assigns higher weight to far min / max outliers) and report the resulting 2 sigma uncertainty in the abstract.

Concerning our (rephrased) claim that "the application of these empirical factors is expected to further improve ... the network conformity ... beyond the scatter among the empirical calibration factors" we cannot provide an explicit proof, as this would require to repeat the calibration for each spectrometer several times. However, in our

opinion it is evident that the application of these factors will further improve the conformity, given that we also have demonstrated the long-term stability of instrumental characteristics for one selected spectrometer. Please note, that the calibration factor for several spectrometers is derived from different calibration runs (sometimes even a year apart), without indication of variable outcome.

**Referee**: "p.2, line 11: "Furthermore these measurements can be directly used to evaluate emissions reductions as demanded by international treaties:" I do not know a study, where such measurements have been used to evaluate emission reductions. Please give a reference or mention that this is a future plan."

**Authors**: Correct, we changed the wording accordingly: "Furthermore, these measurements offer the prospect of being usable for the evaluation of emission reductions as demanded ..."

**Referee**: "p.6, 3.2 It is mentioned that the EM27/SUN spectrometers are operated at a significantly different temperatures. It would be interesting to have a separate assessment of the temperature on the EM27 retrievals."

**Authors**: Our major concern in the context of this paper is the effect of ambient temperature on the spectrometer itself. The demonstrated long term comparison with collocated low-resolution 125HR measurements (this spectrometer is operated in an air-conditioned enforced laboratory container) would reveal any drifts due to ambient temperature. Both setups share the same retrieval setup, so any possible effects of the tropospheric temperature on trace gas results cancel out – as desired, because our study aims at evaluating the hardware performance of COCCON.

**Referee**: "p.8, line 32: Why do HR125 LR and EM27/SUN have a different bias compared to TCCON?"**

**Authors**: The instrument-specific bias of an EM27/SUN spectrometer is the result of instrumental imperfections. We assume that a well-maintained HR125 spectrometer operated at the same spectral resolution as the EM27/SUN is the best realizable approximation of an ideal EM27/SUN spectrometer. The relative difference between the EM27/SUN and the HR125 LR is not at all conspicuous for XCH4. It is higher for XCO2 (0.0014), but still does not exceed the 2 sigma scatter of the calibration factors among the EM27/SUN spectrometers. Please note that all EM27/SUN spectrometers might share a common design feature invoking a common bias with respect to the ideal HR125 LR reference, so we would expect a slightly higher bias when comparing a typical EM27/SUN to the HR125 LR.

**Referee**: "p.9, line 4: "The offset between EM27/SUN and TCCON shows a seasonal variability. Reasons for this are mainly the differences in airmass correction, averaging kernels and retrieval algorithm." Maybe this is discussed in the papers mentioned. However, it is highly important for the network. Therefore the reasons should be (re-) discussed here."

**Authors**: We added an additional citation [Kiel et al., 2016] on this topic discussing in depth a comparable finding of differing seasonal variations due to differing sensitivities of the sensors involved. Please note, that the task of characterization of the emerging network is twofold. The first item are instrumental issues (long-term

stability of instrumental characteristics, instrument-to-instrument variations of instrumental characteristics, stability of the spectrometer when operated under different ambient conditions, especially temperature), the second item is related to the data analysis (preprocessing algorithms, approximations in the radiative transfer code, retrieval strategy, spectroscopic issues, empirical postprocessing steps, etc.). The second – as you mention correctly also important- item is under study in the framework of the ESA project FRM4GHG (http://frm4ghg.aeronomie.be/), for COCCON and other kinds of remote sensing devices and a paper on this topic is in preparation. In the paper under consideration here, we are focusing on the instrumental aspects.

**Referee**: "p.11/12, 4.2: The instrument dependent calibration factors: It would be good to elaborate the discussion about them and have the numbers summarised in a table, e.g. overall biases a) with instrument dependent cal factor (including uncertainties on the cal factor) and b) w/o instrument dependent cal factor."

**Authors**: Table 6 summarizes the biases of the empirical instrument-specific calibration factors and their estimated uncertainties. In our feeling, the provided estimates of the uncertainties as the average absolute deviation are quite conservative (application of the usual Gaussian statistics would assign an additional factor of 1/(SQRT(number of measurement pairs)) when determining the confidence level of the calibration factor). We have expanded the discussion in 4.2 by adding the empirical scatter of the calibration factors from Gaussian statistics (adding more weight to far outliers) and histograms for demonstrating that the statistical distribution of the sample is well-behaved.

**Referee: "p.13 line 9: define how you quantify "scatter""**

**Authors**: We have extended the discussion and now offer two different metrics for the scatter of the empirical instrument-specific calibration factors, based on either the average absolute deviations from the reference spectrometer or on the scatter derived from Gaussian statistics.

**Referee**: "Figure 1: How much information is coming from the apriori in the ILS retrieval? If the constraints are high one can always get a stable ILS. Is it possible to add averaging kernels or similar for the ILS retrieval? At least some explanation should be included in section 3.1."

**Authors**: The problem of an ILS shape not fully determined by the measured spectral scene mentioned by the referee typically occurs in the case of high-resolution spectrometers. The spectral scene used here for the EM27/SUN ILS retrieval is comprised of a large number of water vapor lines generated at ambient pressure. It contains plenty of spectral detail beyond the resolution capability of the EM27/SUN. Therefore, even the extended 40-parameter ILS model offered by LINEFIT could be used without adding significant a-priori information to the ILS solution. However, as the deviations from the nominal ILS of the spectrometers were found to be very small and as we would like to come up with a concise ILS characterization, we applied the simple 2 parameter ILS model.

We added some further information on this in the text: "Due to the fact that the EM27/SUN is equipped with a circular fieldstop aperture, the ILS is nearly nominal.

Therefore, for keeping the treatment concise, we use the simple 2 parameter ILS model offered by LINEFIT."

Referee: "Figure 2: I am not sure if this figure is really needed."

**Authors**: We agree that it might not be of high importance, but in our feeling a figure providing an overview of the complete raw long term dataset is useful, as all the Xgas values discussed in the following are derived from these column data. For this reason, we would like to keep figure 2.

**Referee**: "Figure 4 and 6: The difference in the left figures is difficult to see and contains the same information than the ratio in the figures on the right. Therefore I would leave the difference out in the figures on the left."

Authors: Ok, we have updated figures 4 and 6 accordingly.

Referee: "Figure 10: I would delete this."

**Authors**: We agree that this figure is of marginal importance. However, as referee #2 points out, the figure provides information on the instrument size and illustrates the practical configuration of the side-by-side arrangement. Therefore, we would like to keep it, but we will check in the proof version of the article that its size is appropriate.

**Reference:**

Kiel, M., Hase, F., Blumenstock, T., and Kirner, O. (2016): Comparison of XCO abundances from the Total Carbon Column Observing Network and the Network for the Detection of Atmospheric Composition Change measured in Karlsruhe, Atmospheric Measurement Techniques, 9, 2223–2239, https://doi.org/10.5194/amt-9-2223-2016

Authors' answer to the interactive comments of anonymous referee #2 on "Building the COllaborative Carbon Column Observing Network (COCCON): Long term stability and ensemble performance of the EM27/SUN Fourier transform spectrometer" by Frey et al., Atmos. Meas. Tech. Discuss., amt-2018-146

First of all, we would like to thank the anonymous referee #2 for the help in further improving the current presentment by a thorough assessment with regards of content and the careful technical proofreading resulting in the identification of several imprecisions and typos.

Referee: "Overall comments:

This paper analyzes multi-year analysis of EM27/SUN results compared to TCCON. The long-term performance and stability of the EM27/SUN systems is important to use EM27/SUN results for science analysis and satellite validation. The EM27/SUN systems have potential as lower cost stationary instruments, and for use in shorter term field campaigns since EM27/SUN are easier to move.

I agree with reviewer 1 that column averaging kernels should be shown and compared to TCCON and LR TCCON."

**Authors**: The averaging kernels of the EM27/SUN have already been presented and discussed in comparison to TCCON in the literature [Hedelius et al., 2016]. We have added this information explicitly in the revised version. Please note that the scope of this paper is the characterization of the instrumental performance of the spectrometers used for COCCON.

**Referee**: "The assessment of EM27/SUN results relies on comparisons to a specially processed, modified TCCON dataset, called LR TCCON. LR TCCON is reduced resolution TCCON, with a differently derived ILS, and processed with the PROFFIT software. However, LR TCCON has not itself been validated."

**Authors**: This is true, we are fully aware of this limitation. However, the TCCON LR and EM27/SUN data products have been generated by applying exactly the same processing scheme. The idea behind this approach is to use the TCCON LR dataset to quantify instrument-to-instrument biases and possible instrumental drifts. This approach offers a much higher sensitivity in this regard than a direct comparison with official TCCON products derived from high-res interferograms, because the sensitivities of the reference and device under test are perfectly matched. The fact, that the TCCON LR data product is unvalidated is not harmful in our context, as we are aiming at only a highly sensitive relative comparison (between comparable sensors, comparable in the sense that we expect identical trace gas results if the same atmospheric state is observed).

**Referee**: "Significant differences are seen between EM27/SUN and the full resolution TCCON (shown in Figures 4 and 6) for XCO2 and XCH4. These errors should be quantified in the paper. The errors are seasonally dependent and look to have peakto-peak seasonal errors of about 1 ppm for XCO2 and 20 ppb for XCH4, larger than the TCCON errors compared to aircraft validation (0.4 ppm for XCO2 and 5 ppb for XCO2 for GGG2014 (Wunch, 2015)). Comparisons of EM27/SUN results to LR TCCON are very good. However, LR TCCON has NOT been validated and comparisons of EM27/SUN versus LR TCCON is NOT validation of the EM27/SUN results and does NOT tie EM27/SUN to WMO."

**Authors**: An exhaustive comparison with TCCON will be given in a paper under preparation by the FRM4GHG consortium. We agree that the tying to WMO suffers from a significantly larger uncertainty than the instrument-to-instrument calibration within COCCON. The instrument-to-instrument calibration should be based on the comparison with TCCON LR, only the tying to WMO needs to be done via the official TCCON data products.

Unfortunately, due to higher spectral resolution the TCCON observations have different sensitivity characteristics than COCCON. If the a-priori profile shape assumed by TCCON differs from the truth (its quality might depend on season, and on the current meteorological situation, as demonstrated in the paper for the situation of polar air intrusion), it will give rise to e.g., seasonal differences between TCCON and COCCON of the observed size. Proof of this is given in section 3.5, where a period of polar air intrusion is discussed and in an upcoming FRM4GHG paper.

Following the suggestion of the referee, we have added a short discussion concerning the level of uncertainty with respect to WMO tying of COCCON, which is significantly higher than the internal consistency. This discussion is based on the results provided in tables 3 and 4, which clearly indicate the higher scatter in the EM27/SUN versus TCCON residuals, suggesting a current calibration uncertainty of 0.15% for XCO2 and 0.24% for XCH4 with respect to TCCON.

**Referee**: "In summary, if LR TCCON can be validated versus aircraft/AirCore with similar errors as the standard TCCON, then this paper will set useful limits on EM27/SUN errors. As the paper stands, validation that must be considered is versus the standard TCCON product, which is marginal for satellite validation and on the high side for other uses."

**Authors**: We do not agree to this statement. The choice of TCCON LR product is fully appropriate for demonstrating the level of internal consistency achievable by COCCON. The paper under consideration does not claim to solve the problem of tying the COCCON data to WMO, as the title says, it aims at demonstrating the long-term stability of the EM27/SUN spectrometer, and it investigates the ensemble performance. The construction of a TCCON LR data set is in our opinion the best possible approach for achieving this.

Note that the vertical sensitivity offered by COCCON differs slightly from TCCON, but is not systematically poorer than TCCON (see Hedelius et al., 2016). Therefore, when a similar dataset of in-situ measurements will be exploited for COCCON, the tying to WMO is expected to be of similar quality as for TCCON. Work in this direction, also including AirCore observations, is under progress in the FRM4GHG project.

**Referee**: "Specific comments

Introduction:

The COCCON project should be introduced in the introduction, with the objectives of the COCCON, and who is participating in COCCON, the length of the project (for example)."

Authors: We added the following paragraph in the introduction:

"COCCON is intended to be a lasting framework for creating and maintaining a greenhouse gas observing network based on common instrumental standards and data analysis procedures. Currently, about 18 working groups operating EM27/SUN spectrometers are contributing. We expect that COCCON will become an important supplement of TCCON, as the logistic requirements are low and the spectrometers are simple to operate. It will increase the global density of column-averaged greenhouse gas observations and due to the fact that the spectrometers are portable will especially contribute to the quantification of local sources."

**Referee**: "In the introduction, add in the importance of TCCON for OCO-2 and GOSAT validation, adding a sentence after line 23 something like: "TCCON stations are also the primary validation for OCO-2 (cite https://oco.jpl.nasa.gov/files/ocov2/OCOC22\_SciValPlan\_111005\_ver1\_0\_revA\_final \_signed1.pdf) and validating the satellite observations at different locations is critical for the validation effort (Wunch et al., 2017).""

**Authors: Ok, done**

**Referee**: "A figure showing the TCCON (original and degraded resolution) and EM27 spectral range and radiance would be helpful for the reader, or a reference to a previous paper showing this."

**Authors**: "We added a reference [Hedelius et al., 2016] in section 2.1. The figure contains a TCCON (original resolution) and EM27/SUN spectrum. We refrain from adding a figure in this paper because we think that the additional information from TCCON (degraded resolution) is marginal."

**Referee**: "The spectral ranges and approximate resolution should be given in wavelength in addition to wavenumber. Some scientists are used to wavelength and the translation is not immediately obvious."

**Authors**: "We included this information in section 2.1. For the sake of readability, in the other sections, only the wavenumber notation is given."

**Referee**: "Section 2.2 The description of the HR125 low resolution data set should include the software used to analyze it. I infer it is PROFFIT, but should be stated."

**Authors: Ok, done**

Referee: "Page 5, line 15. Define ILS, modulation efficiency, phase error."

**Authors**: The paper includes a reference to a paper where the procedure of instrumental line shape measurements is explained in detail [Frey et al., 2015].

Additionally we added a reference for a more general description of the used ILS model [Hase et al., 2012].

**Referee**: "Page 5, line 22. How is the phase error calculated – describe or cite a reference. Why is phase error important? What does it affect?"

**Authors**: We included an additional sentence in the manuscript with a reference to the original LINEFIT paper [Hase et al., 1999]. Figure 1 and 2 of this reference illustrates the effect of differing modulation efficiency amplitudes and phase errors on a spectral line.

**Referee**: "The statement on line 7, page 7, "The remaining difference can be attributed to the different measurement heights of the HR125 (112 m) and EM27/SUN (133 m)." This needs to be further explained and quantified. Is it the total column? It would be useful to the reader to have a calculation accounting for the offset."

**Authors**: In this section total columns are discussed. So here it is expected that the total columns differ for instruments at slightly different heights. For an estimate of the ratio the barometric height formula can be utilized. As for this study the main interest lies in the analysis of XCO2 and XCH4, where the height dependency is expected to largely cancel out, we chose not to dwell on the small differences observed in the total columns at different heights.

**Referee**: "Table 2, it would be useful to show the effect on XCO2, etc, which is the key result. The reader looks between columns and thinks it will probably cancel for XCO2 but is not sure."

**Authors**: We agree that this information is vital. We now include the information not only in the text, but also in the caption of the table. Including the information in the table would enlarge the table too far, and we feel it is important to keep the basic information of the total columns.

**Referee**: "Page 8, line 11, "From this higher variability it can be concluded that the airmass dependency in the official TCCON O2 retrieval is higher than for the PROFFIT retrieval, a finding also observed by Gisi et al. (2012)." This statement needs to be modified for clarity to "...higher than for the PROFFIT retrieval on reduced resolution TCCON measurements.""

Authors: We changed the wording accordingly.

**Referee**: "Page 8, line 25. "There are no obvious steps between the EM27/SUN and the HR125 LR data sets so that it can be concluded that the EM27/SUN is stable." The offset versus time needs to be quantified as well. Step functions and slower drift are both important to quantify."

**Authors**: We changed the wording accordingly: "There are no obvious steps and there is no significant drift between..."

**Referee**: "Page 9 line 7. The green line on Fig. 4 shows significant differences between TCCON and EM27, on the order of 1 ppm it looks like. This seasonal cycle

amplitude difference should be quantified. The pink difference (comparison to LR TCCON to EM27) looks very good. As stated in the overall comments, if the difference of EM27/SUN vs. TCCON is larger than the reported TCCON error, then it is important to determine the cause of this difference. PROFFIT should be applied to the full resolution TCCON data, OR GFIT should be applied to the low resolution TCCON data to separate out the PROFFIT/GFIT differences vs. ILS/truncation differences to determine the source of the difference between full-resolution TCCON and LR TCCON. LR TCCON needs to be validated versus aircraft/AirCore before it can be used to validate EM27/SUN."

**Authors**: As we discussed before in this reply and have illustrated in the paper exemplary on the intrusion event seen in March 2016, the differences are mostly due to different sensitivities. In our context of demonstrating the level of long-term stability and ensemble consistency, it is just important to use a common choice for the EM27/SUN and TCCON LR analyses. A comparison of GFIT with PROFFIT for both high- and low-resolution spectra is well beyond the scope of this paper.

**Referee**: "Similar comment for XCH4. In Fig. 6, differences for XCH4 between EM27 and full resolution TCCON look to have seasonal differences of about 20 ppb, which is higher than the TCCON estimated XCH4 error of 5 ppb."

**Authors**: As explained before, this discrepancy simply reveals the smoothing errors inherent in both time series (TCCON and COCCON). The occurrence of larger differences during an episode of a polar air intrusion mentioned in the paper is clearly demonstrating the mechanism.

Only in simple situations – e.g. if one can assume that a certain excess signal is due to a nearby source generating enhanced values in the boundary layer, one can approximately correct for the differing sensitivity characteristics [Wunch et. al, 2011, Hedelius et al., 2017], but in general, when differences in the seasonal cycle are observed, it is not possible to remove the smoothing error without knowledge of the real mixing ratio profile in the atmosphere.

Referee: "Wording/formatting suggestions:

Line 11, suggestion: change "as demanded by" to "as specified by""

Authors: Ok, done

Referee: "Line 16, word suggestion: "Nonetheless" change to "However""

Authors: Ok, done

**Referee**: "Line 20: "However, recently OCO-2 data was used for estimating the source strength of power plants (Nassar et al., 2017)", would reword to emphasize coverage issues, "Recently OCO-2 data was used for estimating the source strength of power plants (Nassar et al., 2017). However, this can only be done for power plants that lie directly under the OCO-2 overpass locations.""

Authors: We rephrased the sentences as suggested.

**Referee**: "Make the dots bigger on the Fig 2-7 legends. It is very hard to tell which dot is blue and which is black in the legend."

Authors: We will change the size of the dots in the legend.

**Referee**: "Page 7 line 11, "Before, a sensitivity study is provided demonstrating the effect of changes in the ILS on the gas retrieval." I think change "Before" to "First"."

Authors: Ok, done

**Referee**: "I see reviewer 1 suggests deleting Fig. 10. However I think Fig. 10 is useful to show the size of the instrument. Perhaps make this figure small."

**Authors**: As stated in the reply to reviewer 1, we will check that the size of the figure is appropriate in the final version of the paper.

1Karlsruhe Institute of Technology (KIT), Institute of Meteorology and Climate Research (IMK-ASF), Karlsruhe, Germany 2Royal Belgian Institute for Space Aeronomy, Brussels, Belgium 3Division of Geological and Planetary Sciences, California Institute of Technology, Pasadena, CA, USA 4Bruker Optics GmbH, Ettlingen, Germany 5Centre for Atmospheric Chemistry, School of Chemistry, University of Wollongong, Wollongong, Australia 6Japan Aerospace Exploration Agency, Tsukuba, Japan 7School of Engineering and Applied Sciences, Harvard University, Cambridge, MA, USA 8Department of Physics and Astronomy, University of Leicester, UK 9National Centre for Earth Observation (NCEO), University of Leicester, UK 10Environmental Sensing and Modeling, Technische Universität München, Munich, Germany 11Universidad National Autonoma de Mexico, Mexico City, Mexico 12National institute for Environmental Studies, Tsukuba, Japan 13Anhui Institute of Optics and Fine Mechanics, Hefei, China 14Institut für Umweltphysik, Universität Heidelberg, Germany 15Botswana International University of Science and Technology, Gaborone, Botswana 16National Institute for Research and Development in Optoelectronics (INOE), Magurele, Romania 17University of Toronto, Toronto, Canada 18Izaña Atmospheric Research Centre (IARC), Meteorological State Agency of Spain (AEMET), Tenerife, Spain 19Laboratoire des sciences du climat et de l'environment, Gif-Sur-Yvette, France 20Environment and Climate Change Canada, Toronto, Canada abefore at: Karlsruhe Institute of Technology (KIT), Institute of Meteorology and Climate Research (IMK-ASF), Karlsruhe, Germany bbefore at: Institut für Physik der Atmosphäre, Deutsches Zentrum für Luft- und Raumfahrt e. V., Oberpfaffenhofen, Germany

Correspondence: M. Frey (m.frey@kit.edu)

Abstract. In a 3.5 year long study, the long term performance of a mobile, solar absorption Bruker EM27/SUN spectrometer, used for greenhouse gases observations, is checked with respect to a co-located reference Bruker IFS 125HR spectrometer, which is part of the Total Carbon Column Observing Network (TCCON). We find that the EM27/SUN is stable on timescales of several years, the drift per year between the EM27/SUN and the official TCCON product is 0.02 ppmv for XCO2 and 0.9

5 ppbv for XCH4, which is within the  $1\sigma$  precision of the comparison, 0.6 ppmv for XCO2 and 4.3 ppbv for XCH4. The bias between the two datasets is 3.9 ppmv for XCO2 and 13.0 ppbv for XCH4. In order to avoid sensitivity dependent artefacts,

the EM27/SUN is also compared to a truncated IFS 125HR dataset derived from full resolution TCCON interferograms. The drift is 0.02 ppmv for  $XCO_2$  and 0.2 ppbv for  $XCH_4$  per year, with  $1\sigma$  precisions of 0.4 ppmv for  $XCO_2$  and 1.4 ppbv for  $XCH_4$ , respectively. The bias between the two datasets is 0.6 ppmv for  $XCO_2$  and 0.5 ppbv for  $XCH_4$ . qualifying it With the presented long term stability, the EM27/SUN qualifies as an useful supplement for the existing TCCON network in

- 5 remote areas. For achieving consistent performance, such an extension requires careful testing of any spectrometers involved by application of common quality assurance measures. One major aim of the COllaborative Carbon Column Observing Network (COCCON) infrastructure is to provide these services to all EM27/SUN operators. In the framework of COCCON development, the performance of an ensemble of 30 EM27/SUN spectrometers was tested and found to be very uniform, enhanced by the centralized inspection performed at the Karlsruhe Institute of Technology prior to deployment. Taking into account measured
- 10 instrumental line shape parameters for each spectrometer, the resulting average bias across the ensemble with respect to the reference EM27/SUN used in the long term study in  $XCO_2$  is 0.20 ppmv, while it is 0.8 ppbv for  $XCH_4$ . The average standard deviation of the ensemble is 0.13 ppmv for  $XCO_2$  and 0.6 ppbv for  $XCH_4$ . In addition to the robust metric based on absolute differences, we calculate the standard deviation among the empirical calibration factors. The resulting  $2\sigma$  uncertainty is 0.6 ppmv for  $XCO_2$  and 2.2 ppbv for  $XCH_4$ . As indicated by the executed long-term study on one device presented here, the
- 15 remaining empirical calibration factor deduced for each individual instrument can be assumed constant over time. Therefore the application of these empirical factors is expected to further improve the EM27/SUN network conformity beyond the raw residual bias scatter among the empirical calibration factors reported above.

**1 Introduction**

Precise measurements of atmospheric abundances of greenhouse gases (GHGs), especially carbon dioxide (CO2) and methane
 (CH4), are of utmost importance for the estimation of emission strengths and flux changes (Olsen and Randerson, 2004).
 Furthermore these measurements can be directly used to evaluate emissions reductions Furthermore, these measurements offer the prospect of being usable for the evaluation of emission reductions as demanded specified by international treaties, e.g. the Paris COP21 agreement (http://unfccc.int/resource/docs/2015/cop21/eng/l09r01.pdf/). The Total Carbon Column Observing Network (TCCON) (Wunch et al., 2011) measures total columns of CO2 and CH4 with reference precision quality. TCCON

- 25 achieves a calibration accuracy with a  $1\sigma$  error of 0.2 ppmv for XCO2 and 2 ppbv for XCH4 and a total uncertainty budget of below 1 ppmv for XCO2 and below 5 ppbv for XCH4, respectively.(Wunch et al., 2010, 2015), however 
[revised manuscript text omitted]